# Growth of lithium-indium dendrites in all-solid-state lithium-based batteries with sulfide electrolytes

Shuting Luo [1,4], Zhenyu Wang[2,4], Xuelei Li[3], Xinyu Liu[2], Haidong Wang[1], Weigang Ma [1], Lianqi Zhang[3], Lingyun Zhu [2,5 ✉] & Xing Zhang [1,5 ✉]

All-solid-state lithium-based batteries with inorganic solid electrolytes are considered a viable option for electrochemical energy storage applications. However, the application of lithium metal is hindered by issues associated with the growth of mossy and dendritic Li morphologies upon prolonged cell cycling and undesired reactions at the electrode/solid electrolyte interface. In this context, alloy materials such as lithium-indium (Li-In) alloys are widely used at the laboratory scale because of their (electro)chemical stability, although no in-depth investigations on their morphological stability have been reported yet. In this work, we report the growth of Li-In dendritic structures when the alloy material is used in combination with a $Li_6PS_5Cl$ solid electrolyte and $Li(Ni_{0.6}Co_{0.2}Mn_{0.2})O_2$ positive electrode active material and cycled at high currents (e.g., $3.8\,mA\,cm^{-2}$) and high cathode loading (e.g., $4\,mAh\,cm^{-2}$). Via ex situ measurements and simulations, we demonstrate that the irregular growth of Li-In dendrites leads to cell short circuits after room-temperature long-term cycling. Furthermore, the difference between Li and Li-In dendrites is investigated and discussed to demonstrate the distinct type of dendrite morphology.

[1] Key Laboratory for Thermal Science and Power Engineering of Ministry of Education, Department of Engineering Mechanics, Tsinghua University, Beijing 100084, China. [2] Guilin Electrical Equipment Scientific Research Institute Co. Ltd., Guilin 541004 Guangxi, China. [3] School of Materials Science and Engineering, Tianjin University of Technology, Tianjin 300384, China. [4] These authors contributed equally: Shuting Luo, Zhenyu Wang. [5] These authors jointly supervised this work: Lingyun Zhu, Xing Zhang. ✉email: zhuly@glesi.com.cn; x-zhang@tsinghua.edu.cn

The thermal instability of conventional lithium-ion batteries (LIBs), which originates from the intrinsic characteristics of liquid electrolytes, causes safety issues and has become a serious impediment to automotive applications. All-solid-state lithium batteries (ASSLBs) using nonflammable solid electrolytes may not only overcome safety concerns in LIBs but also achieve high energy density[1–5]. Lithium metal is recognized as the most attractive choice for anode materials to achieve high energy density due to its low electrochemical potential ($-3.04$ V vs. the standard hydrogen electrode) and high theoretical specific capacity ($3860$ mAh g$^{-1}$). However, it is difficult for pure Li to be applied in sulfide-based ASSLBs in the short term because of the severe interfacial side reactions[6–8] and the growth of Li dendrites[9–11]. Lithium alloys provide an attractive alternative to construct a stable electrolyte-electrode interface that enables long-term cycling for ASSLBs[12]. Lithium alloys can be easily prepared by the solid-state diffusion method at ambient temperature. A number of alloys including Li-Al[13], Li-In[14,15], Li-Si[16], Li-Au[17–19], and Li-Sn[20] have been reported as interlayers or solid solutions for sulfide-based ASSLBs. Generally, the alloy layer or bulk has a higher lithium diffusivity than pure lithium, which is favorable for lithium transport toward the interface; thus, uniform lithium plating can be achieved[21,22]. In addition, the insertion of lithium into other metals can decrease the lithium chemical potential and suppress the electrochemical decomposition of solid sulfide electrolytes (SSEs)[23,24].

Among various lithium alloys, Li-In alloys are particularly popular due to their mechanical ductility and constant redox potential ($0.62$ V vs. Li$^+$/Li) over a wide stoichiometry range[15]. Li-In alloys are usually considered thermodynamically and kinetically stable materials toward SSEs and are widely used in the laboratory for testing the performance of electrolytes or cathodes. Li-In alloys exhibit favorable long-cycling stability toward SSEs in ASSLBs[25–27]. However, most batteries were cycled at low current ($< 1$ mA cm$^{-2}$) and loading ($< 1$ mAh cm$^{-2}$), and it is unclear whether the Li-In alloy anode is still stable toward SSEs at increased current, which is critical for high-power applications of ASSLBs. There have been few investigations to clarify the issue.

In the present work, we cycled a full cell (Li-In|LPSCl|LNO@NCM622) at high current ($3.8$ mA cm$^{-2}$) and high loading ($4$ mAh cm$^{-2}$) to investigate the interface stability between the sulfide electrolyte and Li-In anode. Unexpectedly, the cell presented a short circuit after almost 900 charge/discharge cycles, which is similar to that using a Li metal anode. Combined with scanning electron microscopy (SEM) and energy-dispersive X-ray spectroscopy (EDX) analyses, we observed the growth of Li-In dendrites in the Li$_6$PS$_5$Cl (LPSCl) solid electrolyte, which led to rapid capacity fading and subsequent cell failure. The underlying mechanism for Li-In dendrite growth was revealed by scanning transmission electron microscopy (STEM), electron energy loss spectroscopy (EELS), Raman spectroscopy, X-ray photoelectron spectroscopy (XPS), and ab initio molecular dynamics (AIMD) simulations. The differences between Li and Li-In dendrites in morphology and growth mechanism were also compared.

## Results and discussion

### Cell failure for ASSLBs using Li-In alloy

The purity of the synthesized LPSCl was determined by XRD (Supplementary Fig. 1). The diffraction peaks are well indexed to standard LPSCl. The homogeneity of the electrolyte LPSCl is demonstrated by the SEM image and EDX mapping of P, S, and Cl, as shown in Supplementary Fig. 2. The electrolyte LPSCl has a high ion conductivity of $5.96 \times 10^{-3}$ S cm$^{-1}$ at room temperature (25 °C), as measured by electrochemical impedance spectroscopy (Supplementary Fig. 3). The NCM622 particles were uniformly coated

by LiNbO$_3$ (LNO) with a thickness of ~10 nm, as shown in Supplementary Fig. 4. The cross-sectional SEM image of the prepared Li-In alloy is shown in Supplementary Fig. 5, with the alloy phase circled by blue dotted lines. Due to the favorable mechanical properties of lithium and indium, Li metal easily diffuses into the In matrix and forms an alloy phase under pressure.

Figure 1a shows the long-term cycling of the assembled cell Li-In|LPSCl|LNO@NCM622 at $3.8$ mA cm$^{-2}$ at 25 °C with a high loading of $4$ mAh cm$^{-2}$. The cell maintained a stable cycling capacity and near 100% columbic efficiency during charge-discharge cycling up to 890 cycles (Fig. 1a). However, the capacity started to decline after 891 cycles as shown in the inset, and finally, the discharge capacity decreased to ~0 at the 897th cycle. Figure 1b displays the related charge-discharge voltage profile from the 891th to the 897th cycle of the cell, in which the charge-specific capacity increased gradually while the corresponding discharge specific capacity decreased. At the 897th cycle, the cell was continuously charged with a consistent capacity increase accompanied by a lower voltage increasing rate, as illustrated in Fig. 1c, which indicates the appearance of an internal short circuit and cell failure. These results are similar to those when using a lithium metal anode, which indicates that the Li-In alloy anode is probably unstable toward sulfide electrolytes, especially after long-term cycling at a high current density and high loading. Nine sets of repeated experiments with the same testing conditions were performed for the Li-In|LPSCl|LNO@NCM622 cell for further verification. The long-cycling performance and distribution of cycling life are shown in Supplementary Fig. 6. All the cells presented a short circuit after long-term cycling, with a cell life mainly distributed in the range of 800–1000 cycles, which demonstrates the validity of cell failure in ASSLBs with Li-In anodes.

### Li-In dendrites growth in LPSCl solid electrolyte

To determine the reason for cell failure using the Li-In alloy anode, we carried out ex situ SEM measurements for the cells with different cycling conditions. Figure 2a shows the cross-sectional SEM image for the cell that was rested for 60 days at the open-circuit voltage. The cathode LNO@NCM622, electrolyte LPSCl, and Li-In anode can be clearly distinguished from the SEM images. Owing to the favorable deformability of Li-In alloy, it easily fills the pores and voids at the bottom region of the electrolyte layer when the cell is assembled under a high pressure (760 MPa), which ensures intimate interface contact.

For the cell cycled 100 times (without a short circuit), as shown in Fig. 2b, unlike the resting cell, the Li-In alloy grew into the electrolyte layer by ~30 μm and exhibited a flame shape at the anode-electrolyte interface. We called the Li-In anode that grew into the electrolyte layer Li-In dendrites. For the short-circuited cell with 897 cycles, as shown in Fig. 2c, the Li-In alloy exhibited growth toward the electrolyte interior at ~500 μm, nearly making contact with the cathode. The Li-In alloy almost entirely penetrated the electrolyte layer without a significant anode layer. Due to the growth inhomogeneity of Li-In dendrites at different positions and limited area of observation, Li-In alloy definitely penetrated through the electrolyte at some locations, resulting in a short circuit and cell failure. In the meantime, there was no significant structure or composition change for the cathode and cathode-electrolyte interface, as demonstrated in Supplementary Fig. 7. Therefore, the cell failure of ASSLB using the Li-In anode at a high current was induced by the growth of Li-In dendrites.

To deeply analyze the morphology of Li-In dendrites in the LPSCl electrolyte, SEM and EDX measurements were conducted at various positions for the cell cycled 897 times. The SEM image

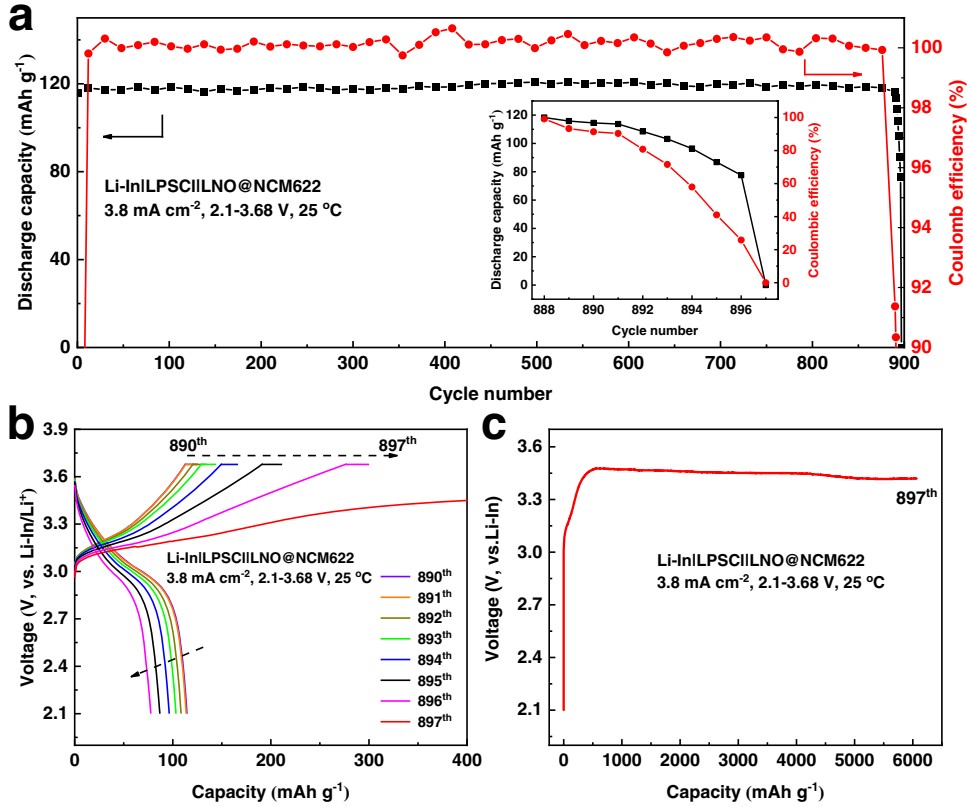

**Fig. 1 Cycling performance of the Li-In|LPSCl|LNO@NCM622 cell. a** Long-term cycling performance at 3.8 mA cm⁻² at 25 °C. The inset shows the variation of discharge capacity and columbic efficiency from the 888th to 897th cycle. **b** Galvanostatic charge-discharge profiles from the 890th to 897th cycle. **c** Galvanostatic charge profile for the 897th cycle.

in the middle region of Li-In dendrites (red boxed area in Fig. 3a) is displayed in Fig. 3b. It is clearly observed that the Li-In dendrites grew densely and laterally in stripes, as further demonstrated by the EDX mapping of indium (Fig. 3c). The elements P, S, and Cl (Fig. 3d–f) were uniformly distributed in the LPSCl electrolyte. The electrolyte particles became smaller than those in the initial state, probably due to the interface reaction. Although the Li-In alloy grew widely in the electrolyte, no cracks or voids were observed. The compactness of the electrolyte indicates that the Li-In dendrites have lower growth stress than Li dendrites without severe structural damage to the electrolyte itself.

Figure 3g shows the morphology of Li-In dendrites in the top region (blue boxed area in Fig. 3a). Different from the streak pattern in the middle region of Li-In dendrites, the morphology displayed a flame shape similar to that of the cell after 100 cycles (Fig. 2b). From the perspective of time and space, it can be inferred that the flame shape is the initial morphology of Li-In dendrites. In the bottom region of Li-In dendrites, the electrolyte was broken into smaller particles <4 μm in diameter, as seen from Fig. 3h (green boxed area). The particle diameter decreased when approaching the bottom. The Li-In dendrites exhibited a network enclosing broken electrolyte particles. Figure 3i shows the SEM image of the transition region between the two morphologies of Li-In dendrites (black boxed area). The upper part presents a streak pattern, which is consistent with the morphology in the middle region of Li-In dendrites. The lower part shows network formation, which is in accord with the bottom layer. In addition, the striped dendrites shown in Fig. 3i are much denser than those in Fig. 3b, and the Li-In alloy occupies a higher proportion in the mixture of the Li-In alloy and electrolyte. Therefore, the damage to the electrolyte caused by dendrites was aggravated as the

cycling process proceeded. To more clearly observe the morphology of Li-In dendrites without the influence of electrolyte, the LPSCl electrolyte and cathode active material NCM622 were removed by a washing method, which was disclosed in the "Methods" section. Figure 3j, k shows SEM images of the Li-In alloy anode from oblique views at low and high magnifications, respectively. It was clear that the Li-In dendrites grew densely and uniformly over a wide region, similar to a matrix wrapping the electrolyte particles.

Based on the above results, it can be concluded that Li-In alloy is unstable toward SSEs when cycled at a high current, even though it exhibits favorable (electro)chemical stability at low current. The formed Li-In dendrites penetrated the solid electrolyte after long-term cycling, eventually resulting in a short circuit and cell failure. From the top to bottom of the Li-In dendrites, the morphology changed from flame shape to stripe and then to the network, presenting a similar evolution in time with increasing cycling number. It should be noted that striped dendrites occupied the majority, with a morphology favorable for reducing the growth rate of Li-In dendrites. More importantly, Li-In dendrites have favorable wettability with electrolyte particles, and the electrolyte layer maintains a high density overall. Therefore, the cell performance was not affected before the electrolyte was penetrated by Li-In dendrites.

Similarly, Li-In dendrites were also observed in nonaqueous liquid-based electrolyte cells under the same testing conditions. The long-term cycling performance of the liquid cell and SEM images of the cycled Li-In anode after dried are shown in Supplementary Figs. 8 and 9, respectively. It can be observed that the surface of Li-In anode was hillocky and porous, while the interior was rather dense. This part of porous structure at the surface can be considered as Li-In dendrites in the liquid cell,

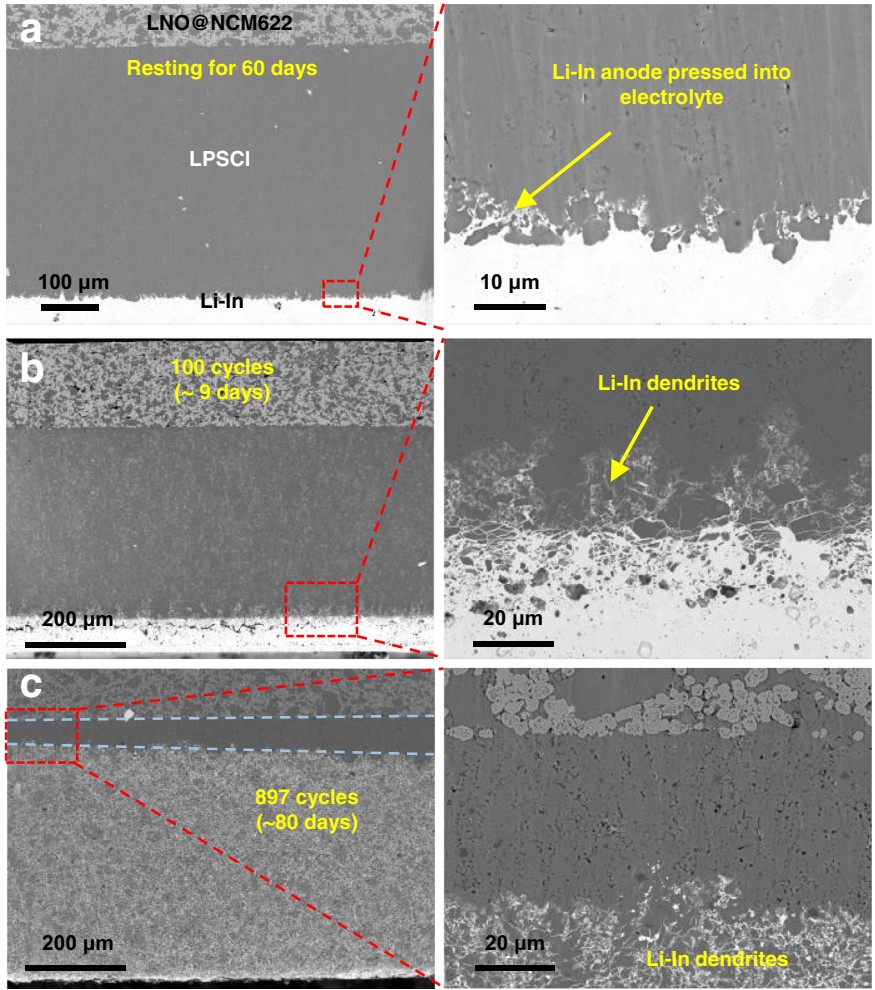

**Fig. 2 Cross-sectional SEM images of Li-In|LPSCl|LNO@NCM622 cells with different cycling numbers. a** The cell after resting for 60 days without cycling. **b** The cell after 100 cycles. **c** The cell after 897 cycles. The cells in (**b**) and (**c**) were cycled at 3.8 mA cm$^{-2}$ under a pressure of 150 MPa within the potential range of 2.1–3.68 V.

which was caused by the repeated expansion and contraction of Li-In anode during cycling. It should be noted that the morphology of Li-In dendrites in liquid cell was different from that in solid cell, which is attributed to the difference of pressure applied on Li-In anode. For the cell with solid electrolyte, it works under a high pressure to remain an intimate interfacial contact. When the Li-In anode expands during charging, there is no enough space to accommodate the expanded Li-In anode. Therefore, it can only grow into the electrolyte along the pores and grain boundaries. On the contrary, for the cell with liquid electrolyte, there is almost no pressure applied on Li-In anode and it can expand freely into the liquid electrolyte. The Li-In dendrites in liquid cell were less likely to induce short circuit due to the obstruction of separator and smoothness of dendrites. But the repeated expansion and contraction during cycling easily cause surface cracking and bulging.

In addition, other solid-state cells with sulfide electrolytes $Li_{10}GeP_2S_{11}$ (LGPS) and $Li_7P_3S_{11}$ (LPS) were also tested. The long-term cycling performance of the cells with LPS and LGPS is shown in Supplementary Fig. 10. Due to the higher ion conductivity of LGPS, it had a higher discharge capacity than LPS in the early stage of cycling. However, the cell with LPS had higher capacity retention owing to the better electrochemical stability. The cells after 300 cycles were stopped and the cross-sectional SEM images of Li-In dendrites in LGPS and LPS are

shown in Supplementary Fig. 11. It can be found that striped Li-In dendrites were presented in both LGPS and LPS electrolytes, which is consistent with the morphology in the LPSCl electrolyte. Moreover, the dendrites in LGPS were much denser than that in LPS and the growth of Li-In dendrites was accelerated in high-reactivity LGPS. Therefore, improving the electrochemical stability of the electrolyte, reducing the porosity of the solid electrolyte and introducing other elements into the alloy are effective means to inhibit the growth of Li-In dendrites.

**Growth mechanism for Li-In dendrites.** Indium metal is commonly considered stable toward sulfide electrolytes, so it is widely used for ASSLB testing. In addition, metal In does not participate in electrochemical cycling in lithium-ion batteries. How does it grow in SSEs? To determine the reason, STEM images in high-angle annular dark field (HAADF) mode and EELS characterization were performed to reveal the growth mechanism of Li-In dendrites. Figure 4a, b shows the STEM-HAADF images of Li-In dendrites in the middle region at low and high magnifications, respectively. It is clearly illustrated that there exists a 15-nm-thick interphase layer at the LPSCl-LiIn interface, which may be composed of compounds such as $Li_2S$, $LiCl$, $In_xS_y$, and $InCl_y$. It is well known that the STEM image contrast can reflect the difference in atomic number at different regions of the sample. The sample with a higher atomic number has a higher brightness in

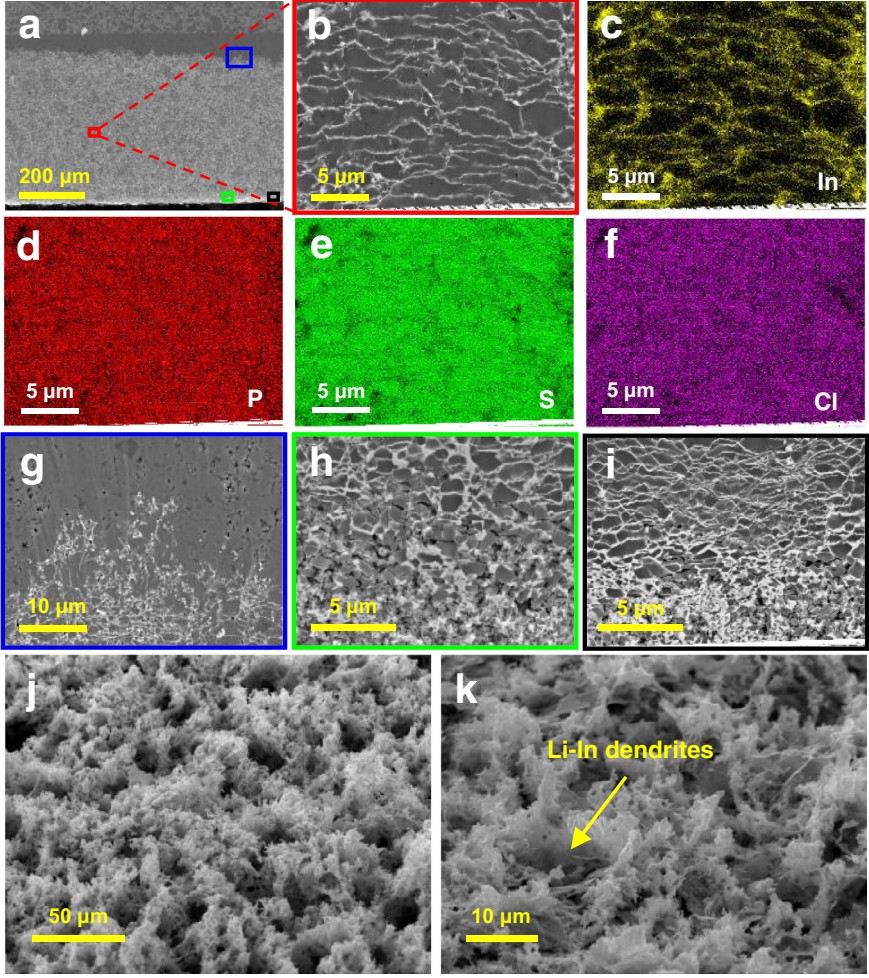

**Fig. 3 SEM images of the Li-In|LPSCl|LNO@NCM622 cell after 897 cycles. a** Cross-sectional SEM images of the cell. **b** SEM image and EDX mapping of **c** In, **d** P, **e** S, and **f** Cl in the middle region of Li-In dendrites. SEM images of Li-In dendrites **g** in the top region, **h** in the bottom region, and **i** in the transition region. SEM images of Li-In dendrites from an oblique view at **j** low magnification and **k** high magnification.

the HAADF image. The average atomic number of the Li-In alloy is 37.5, while that of the electrolyte LPSCl is 10. Therefore, the Li-In dendrite is much brighter than the electrolyte in the HAADF image, which is in accordance with Fig. 4a–c. The brightness of the interphase layer is between the Li-In alloy and electrolyte, which excludes the generation of $Li_2S$ (7.3) and LiCl (10). In addition, the cycling performance shown in Fig. 1a further excludes the formation of $Li_2S$; otherwise, the capacity decreases due to the increased interface impedance[6–8]. Therefore, the interface reaction between Li contained in Li-In alloy and LPSCl can be ignored.

As observed from the STEM-HAADF image and EDX mapping shown in Fig. 4c, d, respectively, there exists a change in elemental composition in the interphase layer. P, S, and Cl in the electrolyte and In exhibit an opposite variation trend at the interphase layer. The interphase layer is mainly composed of indium and sulfur. Some indium-sulfur compounds may be generated at the interphase layer. Therefore, the growth of Li-In dendrites is probably caused by the interaction between metal In and electrolyte LPSCl. Although Li metal is not directly involved in the interface reaction, it may act as a catalyst and facilitate this process. The role of Li metal in the growth of Li-In dendrites deserves further investigation. In addition, the interphase layer has intimate contact with the electrolyte, as observed from Fig. 4b, which facilitates the high compactness of the electrolyte during the growth of Li-In dendrites.

STEM-EELS analysis was conducted to further determine the component of the interphase layer. Figure 4e shows the STEM-HAADF images for Li-In dendrites and corresponding EELS mapping of In and S. The elemental distributions of the interphase layer are more clearly observed from the EELS mapping. The variation trends of In and S in the direction perpendicular to the anode-electrolyte interface agree well with the EDX mapping shown in Fig. 4d. The electron energy loss spectra for Li-In dendrites, LPSCl electrolyte and their interphase layer are shown in Fig. 4f. A new phase that is different from the metal Li-In and LPSCl electrolyte is generated at the interphase layer, which provides direct evidence for the interfacial reaction.

As mentioned above, the interface reaction between Li metal and electrolyte LPSCl was excluded. The interface reaction mainly occurs between metal In and electrolyte LPSCl. Then, first-principles calculations were performed to investigate the chemical reaction of the In-LPSCl interface. The dynamic changes of the LPSCl-In interface were simulated using AIMD at 300 K, and the structural variation was tracked by the radial distribution function (RDF). Figure 5a shows all the formed bonds after AIMD (10 ps) as well as the interface model before AIMD (0 ps). It can be found that many In-S bonds and a small number of In-Cl bonds were formed after MD at the LPSCl-In interface.

Figure 5b shows the RDF evolutions of In-S, In-Cl, and P-S pairs during the simulation and RDFs of In-S and In-Cl for possible generated compounds. The structures of known

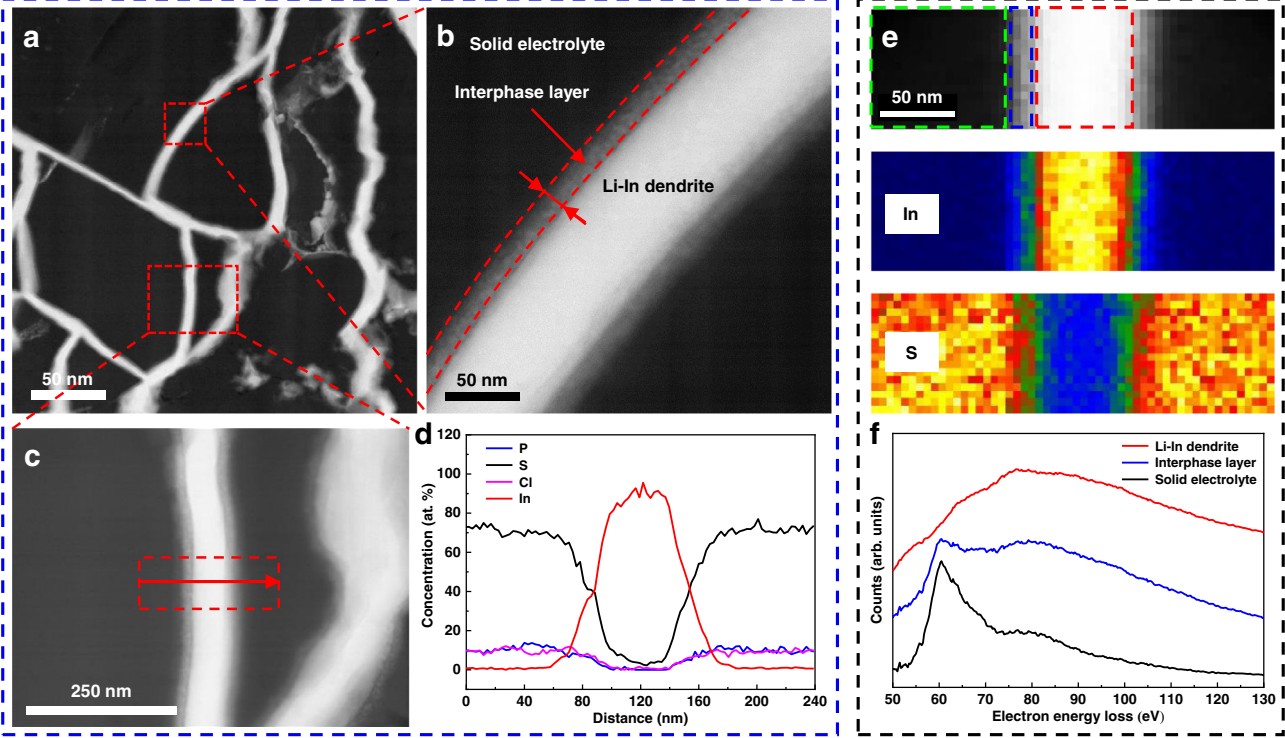

**Fig. 4 STEM-HAADF images and EELS analysis of Li-In dendrites. a** STEM-HAADF images of Li-In dendrites. **b** Magnified Li-In dendrite with interphase layer. **c** Magnified Li-In dendrite used for performing EDX. **d** EDX mapping of P, S, Cl, and In in the direction of the red arrow. **e** STEM-HAADF images of Li-In dendrites and corresponding EELS mapping of In and S. **f** Electron energy loss spectra for Li-In dendrites, LPSCl electrolyte and their interphase layer.

crystalline compounds were obtained from the Materials Project (MP) database. Comparing the RDF evolutions of In-S and In-Cl during the simulation, new peaks of In-S (~2.8 Å) and In-Cl (~2.9 and 3.1 Å) were generated, which means the formation of In-S and In-Cl bonds. The initial formation of In-S and In-Cl bonds occurred at ~1 ps. The intensity of In-S and In-Cl bonds gradually increased and then remained stable during the simulation. The decrease in peak intensity for the P-S RDF indicates the breakage of the P-S bonds. The changes in Li-P, Li-S, and Li-Cl bonds for electrolyte LPSCl during the simulation are illustrated in Supplementary Fig. 12. The brightness of Li-P, Li-S, and Li-Cl gradually became weak with time, indicating the decomposition of electrolyte LPSCl.

To determine the product, we searched the MP database for stable compounds that may be generated by In, S, and Cl. It can be observed that the newly generated peaks match well with $In_2S_3$, InS, InCl, $InCl_3$, etc. Due to the low content of Cl in the LPSCl electrolyte, the amount of $InCl_y$ was much less than that of $In_xS_y$, as reflected by the number of In-S bonds and In-Cl bonds in Fig. 5a. Therefore, $In_xS_y$ is possibly the main reaction product.

To further confirm the occurrence of the interface reaction and determine the generated product, a contact experiment was performed with pure In foil in contact with the electrolyte LPSCl for 7 days. Then, the electrolyte LPSCl was removed by the washing method illustrated in the "Methods" section. Raman spectroscopy and XPS measurements and analyses were carried out to investigate the surface of In foil contacting LPSCl. Moreover, fresh In foil without contact was analyzed for contrast. Figure 5c shows the SEM images and Raman spectra of the fresh In foil and contacted In foil. Many hillocks appeared on the surface of the contacted In foil due to intimate combination of the electrolyte layer and In foil under high pressure. Comparing the Raman spectra for the surface of In foils, a new peak at 300 cm$^{-1}$ was observed for the contacted In, which matched well with the

Raman spectrum of standard material $In_2S_3$, demonstrating the formation of $In_2S_3$. XPS depth analysis for the contacted In foil is shown in Fig. 5d. Notably, In 3d peaks and S 2p peaks of $In_2S_3$ were detected at the surface of the In foil, consistent with the Raman spectra. With increasing etching depth, the peaks of In 3d sifted to higher binding energy, indicating a valence transition from $In^{3+}$ to $In^0$. Meanwhile, the S 2p peaks gradually decreased in intensity, suggesting penetration of the interphase layer. As seen from the atomic concentrations of In, S, and Cl from the depth profiles in Fig. 5d, the interphase layer is mainly composed of In and S, with no significant variation in the Cl content detected. Therefore, $In_2S_3$ is the reaction product between metal In and electrolyte LPSCl.

The influence of the interface reaction on cell resistance was investigated by electrochemical impedance spectroscopy (EIS) measurements. Supplementary Fig. 13 shows the evolution of the electrochemical impedance spectrum and corresponding internal resistance of the In|LPSCl|In cell after 144 h. It can be clearly found that the internal resistance first decreases and then remains stable. The reduction in internal resistance is attributed to the improved interface contact caused by the interfacial reaction. The interphase layer has favorable wettability with the electrolyte, as indicated in Fig. 4b, c. The subsequently unchanged resistance represents the formation of a stable interphase layer. Therefore, the interface reaction between In and LPSCl does not induce increased internal resistance, which agrees well with the favorable cycling performance shown in Fig. 1a.

AIMD simulations and Raman and XPS analyses demonstrate that chemical side reactions occur at the LPSCl-In interface and that the generation of $In_2S_3$ is thermodynamically favorable. However, whether the growth of Li-In dendrites can continue depends not only on thermodynamic favorability but also on kinetic feasibility. The growth rate of Li-In dendrites is closely related to the cycling current and cathode loading. When the cell

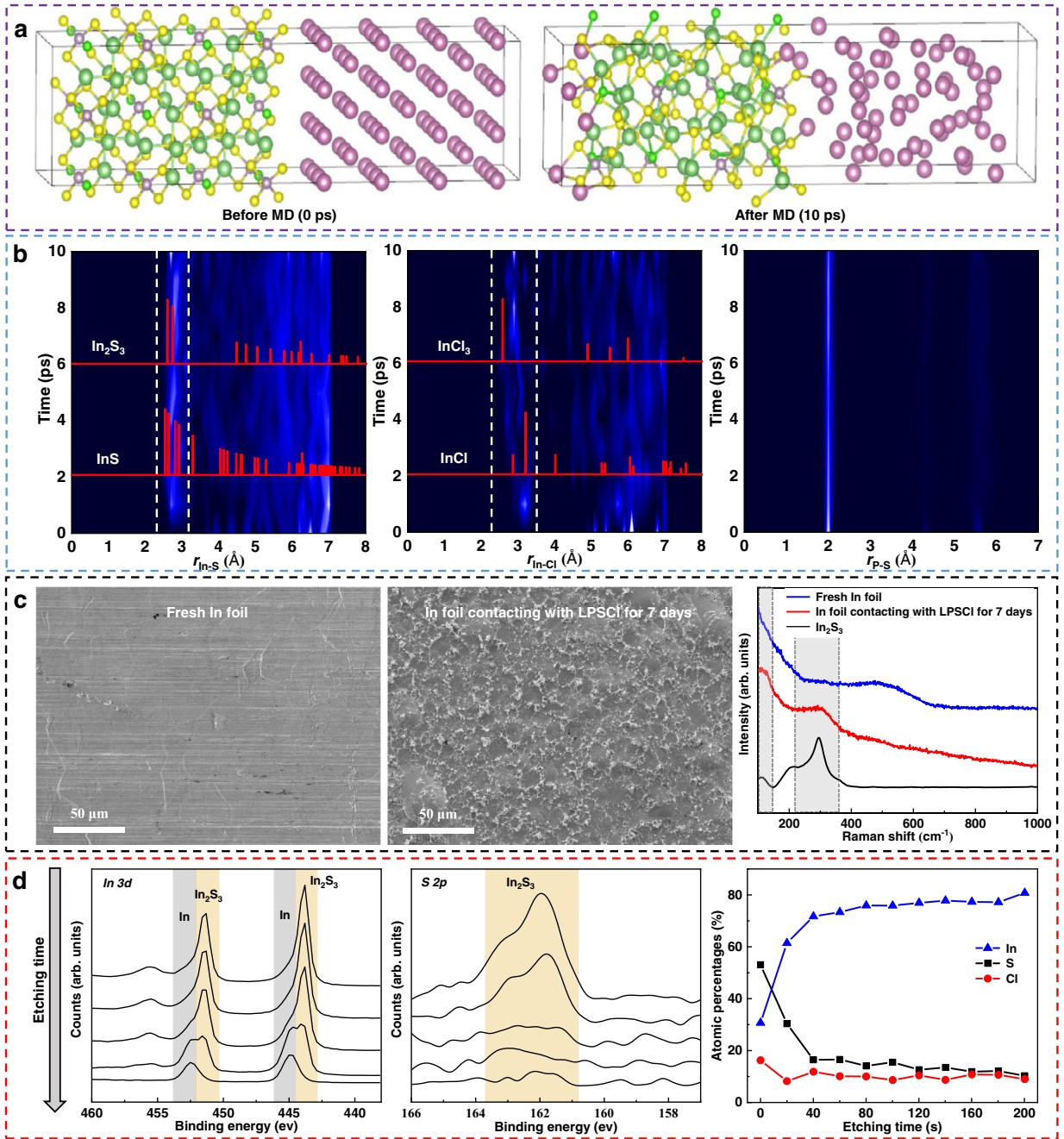

**Fig. 5 AIMD calculations and Raman and XPS analysis of the LPSCl-In interface. a** LPSCl-In interface models before MD (0 ps) and after MD (10 ps). **b** RDF evolutions of In-S, In-Cl, and P-S for the LPSCl-In interface during the simulation and RDFs of In-S and In-Cl for reference materials. **c** SEM images and Raman spectra of fresh In foil and In foil in contact with LPSCl for 7 days. **d** XPS depth profiling analysis of In foil in contact with LPSCl for 7 days.

is cycled at a high current, large amounts of lithium ions enter the indium matrix during charging, which induces significant volume expansion of the dendrite tip. Grain boundaries and pores are the preferential expansion channels because they have the least resistance. The experimental results shown in Supplementary Figs. 14 and 15 demonstrate that the growth rate of Li-In dendrites is positively associated with the cycling current. In addition, the high loading increases the single expansion time of the indium matrix, accelerating the growth rate of Li-in dendrites. The newly produced $In_2S_3$ at the interface improves the wettability behavior between Li-In and LPSCl particles, which results in an intimate contact interface and matrix dendrite structure. Therefore, the

formed Li-In dendrites fill the grain boundaries and pores like liquid and tightly enclose the electrolyte particles, which promotes the structural stability of the electrolyte and even improves the overall density. High loading and high current are necessary conditions for the growth of Li-In dendrites. Specific energy (Wh kg$^{-1}$) and power (W kg$^{-1}$) are positively correlated with the cathode loading and current density, respectively. Supplementary Fig. 16 shows the Ragone plots for the cells reported in the literature which employ sulfide electrolytes and are cycled at room temperature. The value of the present work is in the upper right region of the Ragone plots, indicating high specific energy and power. Therefore, the growth of Li-In dendrites is more likely to

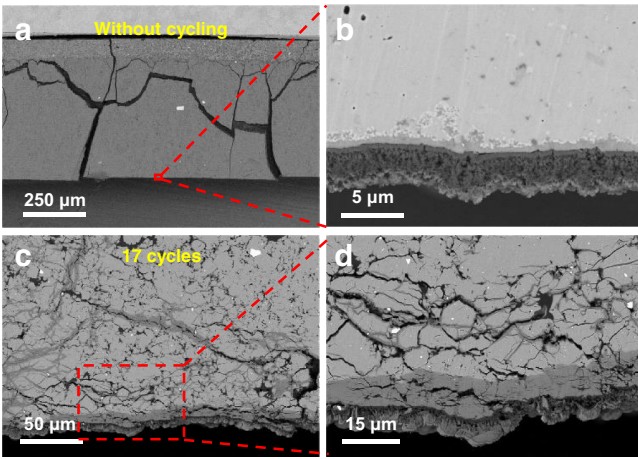

**Fig. 6 Cross-sectional SEM images of the Li|LPSCl|LNO@NCM622 cell.** SEM images of **a** the overall cross section and **b** magnified Li-LPSCl interface without cycling. SEM images of the cell after 17 cycles at **c** low magnification and **d** high magnification.

occur in high-energy and high-power batteries, which should be given attention in the testing and application of ASSLBs.

**Comparison between Li-In dendrites and Li dendrites.** Lithium dendrites have been widely investigated in recent decades; however, the formation of Li-In dendrites was not previously reported. To determine the difference between Li and Li-In dendrites, we also carried out electrochemical measurements and SEM observations for the Li|LPSCl|LNO@NCM622 cells. The cycling performance and galvanostatic charge-discharge profiles are shown in Supplementary Fig. 17. Compared with the cell using the Li-In anode, the Li anode cell has a shorter cycling life (17 cycles) and a much lower charge/discharge current ($0.3 \, \mathrm{mA \, cm^{-2}}$). A similar short circuit occurs in the last few cycles due to the growth of Li dendrites. The charge capacity gradually increases, while the discharge capacity and coulombic efficiency simultaneously decrease and finally drop to zero.

Figure 6a, b present the cross-sectional SEM images in backscattered electron (BSE) mode for the fresh Li|LPSCl| LNO@NCM622 cell without cycling. The cracks are caused by cell disassembly because no binder is added to the electrode and electrolyte. The magnified image of the boxed area includes a part of the LPSCl electrolyte adjacent to the Li anode. The interphase layer at the interface is different from the Li anode and LPSCl electrolyte observed from the high-contrast BSE image, indicating a severe interfacial side reaction between Li and LPSCl. After 17 cycles, as shown in Fig. 6c, d, the growth of Li dendrites is much sharper than that of Li-In dendrites and exhibits a different growth morphology. Li dendrites not only grow along the grain boundaries and pores but also damage the electrolyte structure and induce severe crack propagation, which results in a rapid short circuit within several cycles. Herein, it should be noted that the thin Li anode (~5 μm) shown in Fig. 6 is also induced by cell disassembly. The lithium anode was not completely stripped from the current collector and adhered to the stainless steel. The thickness of the Li anode used in the testing was ~100 μm.

Compared with the measurements and analyses of the Li-In dendrites, we can find that there are significant differences between Li-In and Li dendrites. First, they have different growth morphologies. Lithium dendrites grew vertically, perpendicular to the anode-electrolyte interface. Lithium-indium dendrites grew laterally in stripes, much denser and much more uniform than Li dendrites. The reason is that the growth of Li dendrites is induced by the nonuniform Li deposition that prefers to form whiskers,

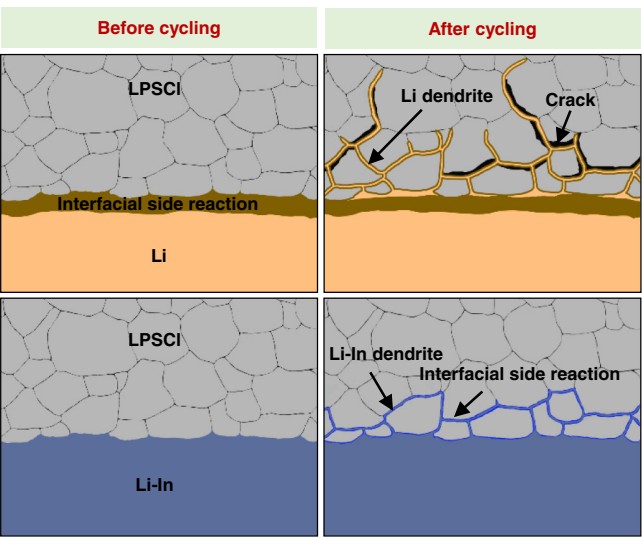

**Fig. 7 Schematic diagram of LPSCl-Li and LPSCl-LiIn interface evolution in cells before and after cycling.** The interfacial side reaction between LPSCl and Li is much severe than that of LPSCl-LiIn interface before cycling. Li dendrites vertically grow in the electrolyte layer accompanied by many cracks during cycling, while Li-In dendrites laterally grow along the pores and grain boundaries with high compactness of electrolyte layer.

while Li-In dendrites are caused by the volume expansion and slight interface reaction that prefers to fill the grain boundaries and pores. Second, these dendrites have different contact behaviors with the solid electrolyte. The growth of Li dendrites causes severe crack propagation due to the stress concentration and high reactivity, leading to a loose and porous electrolyte structure. However, for the metal Li-In anode, the formation of a thin interphase layer (15 nm) due to the slight interfacial reaction enables a favorable contact between the LPSCl-LiIn interface, allowing a dense electrolyte structure to be achieved. A schematic diagram of LPSCl-Li and LPSCl-LiIn interface evolution in cells before and after cycling is shown in Fig. 7.

In summary, we report the growth of Li-In dendrites in sulfide electrolytes for high-energy and high-power all-solid-state lithium batteries. The dendrite morphology was characterized by SEM-EDX analysis, and the growth mechanism was revealed by STEM-EELS, Raman spectroscopy, XPS, and AIMD simulations. Our findings suggest that metal In is thermodynamically and kinetically unstable toward sulfide electrolytes when the cell is cycled at high current and high loading. The accompanying volume change and slight interfacial reaction induce the growth of Li-In dendrites enclosing electrolyte particles, eventually leading to short circuits and cell failure after a long cycle. Compared with vertically growing Li dendrites (Fig. 6b), laterally striped Li-In dendrites (Fig. 3b) are favorable for reducing the growth rate of dendrites and alleviating structural damage to sulfide electrolytes. The growth of Li-In dendrites can be inhibited by improving the electrochemical stability of the metal electrode/solid electrolyte and reducing the porosity of the electrolyte. Our investigation reveals the failure mechanism of ASSLBs using Li-In anodes and provides valuable insight into Li-In dendrites, which will provide important guidance for benchmarking laboratory-scale solid-state battery studies and the development of alloy anodes.

## Methods
**Material synthesis**. The sulfide solid electrolyte LPSCl was produced by GLESI, China. Li₂S (> 99.9% purity, Alfa Aesar), P₂S₅ (>99% purity, Alfa Aesar), and LiCl (> 99.9% purity, Alfa Aesar) were first mixed in an appropriate molar ratio. The mixture was then placed in a ZrO₂ pot containing a ZrO₂ ball to be mechanically milled using a planetary ball milling apparatus at 500 r.p.m. for 30 h. Following the

ball milling procedure, the mixture was heated at 550 °C for 6 h under an argon (Ar) atmosphere with $H_2O$ and $O_2 < 0.1$ ppm.

LiNbO$_3$-coated LiNi$_{0.6}$Co$_{0.2}$Mn$_{0.2}$O$_2$ (LNO@NCM622) was provided by GLESI, China. By means of fluidized bed technology, an LNO coating layer was formed with an ethanol solution of Li and Nb. Metal Li and niobium ethoxide (>99.9% purity, Alfa Aesar; molar ratio, Li:Nb = 1:1) were dissolved in anhydrous ethanol (Kanto Chemical) under an Ar atmosphere. The solution was sprayed onto NCM622 particles at a spraying rate of 2 g min$^{-1}$ by a rolling fluidized coating machine (Powrex, MP-01). LNO@NCM622 was finally heated at 400 °C for 60 min under $O_2$. The coating layer thickness can be controlled by the spraying duration. A lithium-indium (Li-In) alloy was prepared by the solid-state diffusion method. A piece of lithium (~30-μm thickness) and a piece of indium (~100-μm thickness) with a weight ratio of 2:98 were pressed together at 760 MPa. Due to the favorable mechanical properties of lithium and indium, Li metal easily diffuses into the In matrix, and a Li-In alloy can be obtained. Then the obtained Li-In alloy was cut into disks with a diameter of 10 mm and thickness of ~40 μm to assemble cells. The electrodes and electrolytes were prepared in the Ar atmosphere with $H_2O$ and $O_2 < 0.1$ ppm.

**Fabrication of an all-solid-state lithium cells.** As-synthesized LNO@NCM622 powder was manually mixed with LPSCl electrolyte powder using a quartz mortar for 20 min in a mass ratio of 70:30, and the obtained composite was used as the cathode for the full cell. Li-In alloy foil was used as anode. Cells were assembled as follows: 70 mg LPSCl electrolyte powder was pressed under 150 MPa into an electrolyte layer with a diameter of 10 mm. Then, 40 mg composite cathode powder was uniformly spread on one side of the electrolyte layer, and stainless steel (SS) foil was placed on the composite cathode layer. The thickness of the SS foil was 200 μm, which was produced by Xinyu Stainless Steel Material Company. Next, Li-In alloy (~25 mg) was placed on the other side of the electrolyte layer with SS foil as the collector. Finally, all the components were compressed together under 760 MPa to form a full cell. The geometry of the full cell was similar to that of a coin with a diameter of 10 mm and a thickness of 600 μm. The pressure applied on the cell during cycling was ~150 MPa. The cells with other sulfide electrolytes (LGPS and LPS) were assembled using the same method. The weight of the electrodes and electrolyte, the cycling pressure, and the testing conditions were consistent with those of the Li-In|LPSCl|LNO@NCM622 cell.

For the reference cell Li|LPSCl|LNO@NCM622, the Li-In anode was replaced by Li metal (~100-μm thickness), and the weight of the composite cathode was reduced to 10 mg. The pressure was changed from 760 to 100 MPa, with other preparation conditions remaining consistent. For the liquid cell, 1 M LiPF$_6$ in 1:1 vol/vol EC/DEC with FEC additive was used as the liquid electrolyte. The weight of the cathode active material in the liquid cell was consistent with that in the all-solid-state cells. All the cells were fabricated in an Ar-filled glovebox with $H_2O$ and $O_2 < 0.1$ ppm.

For the assembled Li-In|LPSCl|LNO@NCM622 cell with a diameter of 10 mm, the cathode loading was 4 mAh cm$^{-2}$ based on the discharge capacity of the initial cycle. For the Li-In anode (25 mg), the amounts of metal In and metal Li were 24.5 mg and 0.5 mg, respectively. The capacity of metal In is 6 mAh cm$^{-2}$, calculated from the theoretical specific capacity of 194 mAh g$^{-1}$. Li metal was used to supplement the lithium lost by the cathode during cycling; thus, it is not included in the anode capacity. Therefore, the theoretical capacity of the Li-In anode is 6 mAh cm$^{-2}$, which is higher than that of the cathode. Therefore, a 67% theoretical capacity of the Li-In anode was used per cycle.

**Materials characterization.** X-ray diffraction (XRD, Rigaku D/MAX-500, Japan) of the LPSCl electrolyte was performed by using Cu Kα radiation in the 2θ range of 10–65° with a step size of 0.02°. Scanning electron microscopy (SEM) equipped with energy-dispersive X-ray spectroscopy (EDX) (JEOL, JSM-7900F, Japan) was used to analyze the cross-sectional morphology of the cell. The sample cross-section was polished using a cross-section polisher (JEOL, IB-19520CCP, Japan) before SEM-EDX analysis. The microstructure of the tested sample was characterized by cryo-scanning transmission electron microscopy (cryo-STEM, HD2700, Hitachi) at −100 °C. The elemental distribution was determined via EDX (XMAXN 100TLE, Oxford), and component analysis was performed using electron energy loss spectroscopy (Enfinlum, Gatan). STEM specimens were obtained by thinning the electrolyte layer with dual-beam focused ion beam equipment (FIB, NB5000, Hitachi) operated at 2–30 kV. A special compatible holder was used to enable the sample to be directly transferred from the FIB equipment to cryo-STEM without exposure to air. The electrolyte and cathode were washed as follows: a part of the dismantled specimen of the cell was soaked and washed in deionized water for 2 h and then dried for 24 h. By which, the LPSCl electrolyte can be removed via chemical reaction. The remaining active material NCM622 could not maintain a complete structure and could be washed away by deionized water.

**Electrochemical measurements.** The conductivity of LPSCl was measured on an SS|LPSCl|SS cell using an electrochemical workstation (EIS, Bio-Logic VSP-300) over a frequency range of 1 Hz–6 MHz at 25 °C with a potentiostatic signal. The number of data points was 10 per decade, and the amplitude of the signal was 10 mV. The excitation signal mode was a single sine at a quasi-stationary potential.

For long-term cycling, the Li-In|LPSCl|LNO@NCM622 cell was charged to 3.68 V (vs. Li-In/Li$^+$, 4.3 V vs. Li/Li$^+$) in constant current mode (3.8 mA cm$^{-2}$) and then charged in constant voltage mode (3.68 V) for 15 min. Next, the cell was discharged at 3.8 mA cm$^{-2}$ to 2.1 V (vs. Li-In/Li$^+$, 2.72 V vs. Li/Li$^+$). The average charge and discharge voltage were 3.4 and 3.1 V, respectively. The cell with the Li anode was discharged at a constant current mode (0.3 mA cm$^{-2}$) in the potential range of 2.7–4 V. The liquid cell was cycled with the same testing conditions as the all-solid-state cells. After cycling, the liquid cell was disassembled, and the Li-In anode was dried at 80 °C for 40 h to remove the liquid electrolyte to further observe the surface and cross-sectional morphology. All electrochemical measurements were performed at 25 °C in an Ar atmosphere with $H_2O$ and $O_2 < 0.1$ ppm.

**AIMD calculations.** AIMD simulations were performed in the Vienna Ab initio Simulation Package (VASP) with the projector augmented wave (PAW) method. The exchange-correlation potential was described by the generalized gradient approximation (GGA) function, which was parameterized with the Perdew–Burke–Ernzerhof (PBE) functional. A 500 eV cutoff energy for plane waves was set, and a $1 \times 1 \times 1$ K-point mesh in the first Brillouin zone was used as the large atom number (>250) of the interfacial models. An NVT ensemble based on a Nosé thermostat and a time step of 2 fs was adopted in the AIMD simulations.

**Reporting summary.** Further information on research design is available in the Nature Research Reporting Summary linked to this article.

## Data availability
The source data used in this study are available in the Figshare database (https://doi.org/10.6084/m9.figshare.15021981) (ref. [28]).

## Code availability
The computer code used in this study is available in the Figshare database (https://doi.org/10.6084/m9.figshare.15021981) (ref. [28]).

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

## Acknowledgements

This work was supported by the National Key Research and Development Program of China (2018YFB0104300), the National Natural Science Foundation of China (Grant Nos. 51827807 and 51636002), the Science and Technology Major Project of China National Machinery Industry Corporation (SINOMAST-ZDZX-2019-04) and the Special Funds for Innovation Driven Development of Guangxi Zhuang Autonomous Region (AA17204061).

## Author contributions

X.Z. and L.Z. proposed the topic of this perspective. S.L. and Z.W. designed the experimental plan and performed the literature search. X.L. (Xuelei Li) designed the figures. X.L. (Xinyu Liu) helped with the EIS and XRD measurements. H.W., W.M., and L.Z. made contributions to the scientific discussion. S.L. analyzed the experimental results and wrote the manuscript.

## Competing interests

The authors declare no competing interests.
