## [Peer Review File · Nature Communications]

REVIEWER COMMENTS

Reviewer #1 (Remarks to the Author):

On this manuscript the prevalence of Li-In alloy dendrites in sulfide electrolyte-based solid-state batteries is described. The authors combine various microscopic and spectroscopic methods to show that Li-In dendrites form upon cycling such solid-state batteries. Although the use of Li-In alloy as an anode in solid-state batteries is unlikely to prevail for commercialization of solid-state batteries (due to the high cost of In), the results presented in this manuscript provide crucial fundamental insights that can help benchmark lab-scale solid-state battery studies, especially at high cycling rates.

The presented results are self-consistent and tell a reasonable story that warrants publication after the authors clarify some concepts and provide feedback to the following comments:

1. The phase purity of the chloride argyrodite solid electrolyte (Li₆PS₅Cl) was tested against an entirely different material (Li₇PS₆). Even if the structure of both compounds is the same, there is no reason to not index/analyze the pattern of the solid electrolyte with the target phase. Seems a bit odd.
2. Although the reported room-temperature ionic conductivity of the chloride argyrodite is within previously reported values (see e.g. Ohno et al. ACS Energy Lett, 2020, 5, 9, 910), the electrical equivalent circuit shown in Figure S2 is invalid for analyzing such a spectrum and is unlikely to have given such a value. Based on the inset shown in Figure S2, extrapolating the intercept at the real impedance axis (it should be ca. 70–75 Ohms), and then using the pellet dimensions to determine the ionic conductivity should be sufficient and more correct.
3. In Figure 3, the authors show that there are In-rich regions (close to the initial anode-solid electrolyte interface) and (regular?) Li-In dendrites that have grown up to the opposite cathode-solid electrolyte interface. It is however unclear to me how were these In-rich regions identified, and more importantly how do these compositional changes in the alloy affect the cell potential. Also what exactly is the mechanism for these spatially-resolved compositional changes in the alloy dendrites? Is the diffusivity of Li hindered due to the formation of vacancies as was also demonstrated by the Janek group with the LiMg alloy in contact with garnet electrolytes? (see: Krauskopf et al. Adv. Energy Mater. 2019, 9, 1902568). Since the authors are talking about growth mechanisms, there could/should be more describing the fundamentals of this process.
4. In Figure 4, it is unclear if the dendrites imaged correspond to the In-rich phase or the “regular” Li-In dendrites. Also, is the composition of the interphase layer dependent on whether it is an In-rich dendrite or a “regular” Li-In dendrite?
5. I am not a theoretical expert, so the following comments should be taken with that statement in mind:
 - a. Why is the simulation done only considering pure indium as the anode and not an Li-In alloy?
 - b. The authors mention that they used the crystalline compounds InS and InCl to calculate the RDFs, however, I am not convinced that these compounds actually exist, especially the indium sulfide (In¹⁺ can be stable and makes InCl more reasonable). Can the authors comment a bit on what structures were actually considered for this analysis?
 - c. What exactly happens to the Li and P atoms once In-S and In-Cl bonds start forming?

Reviewer #2 (Remarks to the Author):

Strong points: If the results are confirmed, the finding is indeed interesting. At the end, the study shows that a much longer cycle life can be obtained by using alloys instead of pure Li metal.

Limitations:

Unfortunately I see some limitations for this paper which makes it difficult to judge on the relevance of the results. Claims are not fully supported at this stage.

A) Information on the Li-In electrode is very vague. It says the total weight is approx. 25 g and the Li:In ratio 2:98. But what are the relevant values? Electrode thickness? Capacity of Li of this electrode is not mentioned at all. Is it also 4 mAh/cm²? How was such little Li applied?

B) Same thing for the Li electrode, which is apparently only 5 μm thick. How was this done and confirmed?

C) There is not a real proof for that the dendrites are really a Li-In alloy. Of course, this is not easy but since there is no evidence it is only a hypothesis. Could it be not simply formation of InS along the grain boundary as also suggested from theory? The authors could check this by simply making another cell and simply let it rest for long time (without cycling). Maybe the In reacts with the solid electrolyte?

D) If it is true that Li-In dendrites grow, they should also do this in cells with liquid electrolyte. Did the authors consider doing the same experiment in cells with liquid electrolyte? This would support the claim that Li-In alloys can form dendrites.

E) Studies on dendrites are highly erratic, so one cell may fail after 800 cycles, the other one at 100 cycles. It is clear that long-time measurements are time consuming but the authors should at least comment on how many experiments with cell failure were done.

F) Authors claim that the dendrites grow only along grain boundaries but there is no information on whether the solid electrolyte is really dense. SEM is not sufficient here. Did the authors try to determine the density of the pellet? Does it correspond to the theoretical density? Otherwise the dendrites might grow along the pores.

G) Figure 2 shows already lot of cracks/pores before cycling. Is it then reasonable to have the discussion on crack formation in case of lithium cells and not crack formation for the Li-In cells?

H) Authors state that they remove the electrolyte and the cathode material using deionized water by a chemical reaction. It is impossible to dissolve cathode material by water. It is hard to understand what the authors.

I) Interface: Authors state that InS forms, but what about Li₂S?

J) Page 10: Statement on high/low current. The paper only shows results at one current density. I can not follow the discussion at this point.

K) I find the title quite misleading as it does not highlight the major finding of the paper. Besides, "High energy and Power" is not addressed in this paper, so why mentioning this? With a 25 g In electrode and such a very thick solid electrolyte the cell will have very low energy and power. Did the authors calculate this?

There are smaller aspects that should be checked. Spelling mistakes like "columbic efficiency" (should

be "Coulomb efficiency"). What is the difference between a "whisker" and a "dendrite"?

Reviewer #3 (Remarks to the Author):

In this paper, the author observed the dendrite growth in Li-In alloys and analyzed its mechanism. The findings on dendrite growth of Li-In contain a certain amount of novelty. However, the reviewer thinks that the Li-In anode is not suitable as an anode material for high energy all-solid-state batteries. In addition, it is stated that the interface between the Li-In alloy and the electrolyte was found to be unstable in this paper, but considering that the Li₆PS₅Cl is unstable under the redox potential of Li-In (0.62 V vs. Li⁺/Li) from theoretical calculation (G. Ceder et al., Chem. Mater. 2016, 28, 266–273), it is obvious that the interface between the Li-In alloy and the electrolyte is unstable. In summary, although this paper provides new insights into dendrite growth in Li-In, its impact on the development of high-energy all-solid-state batteries is small, and the reviewers recommend that the authors submit this paper to a more specific journal. More specific comments are described below.

Major comments:

(1) Page 3, Line 10

Please write the correct composition of LPSCl when it first appears.

(2) Page 3, Line 19-20 and Figure S1

As there are a few peaks that cannot be indexed Li₇PS₆ in Figure S1 around 34 and 47°, the prepared LPSCl is not single phase.

(3) Page 4, Line 1-4 and Figure S4

Although the author mentioned that "the formed alloy phase is uniformly distributed in indium matrix", the cross-sectional SEM image of the Li-In alloy is not uniform.

(4) Page 10, Line 1-3 and Figure 4

As mentioned above, the Li₆PS₅Cl is unstable under the redox potential of Li-In (0.62 V vs. Li⁺/Li) from theoretical calculation (G. Ceder et al., Chem. Mater. 2016, 28, 266–273), it is normal that the Li-In alloy is unstable towards Li₆PS₅Cl solid electrolyte.

(5) Figure 7

The author should reflect on the interfacial side reaction between Li-In alloy and Li₆PS₅Cl into the schematic diagram.

Minor comments:

(6) Page 19, Line 2 and Page 20, Line 3

The font of °C is different from the other characters.

(7) Page 19, Line 5

Mpa should be revised to be MPa.

RESPONSE TO REFEREES

We are very grateful to the referees for their thoughtful suggestions and careful reviews on our manuscript. According to the reviewers' suggestions, we have revised our manuscript. The detailed changes corresponding to the reviewers' comments are listed below:

For the first reviewer:

#Q1.1: The phase purity of the chloride argyrodite solid electrolyte ($\text{Li}_6\text{PS}_5\text{Cl}$) was tested against an entirely different material (Li_7PS_6). Even if the structure of both compounds is the same, there is no reason to not index/analyze the pattern of the solid electrolyte with the target phase. Seems a bit odd.

#A1.1: Thanks for your comments. After careful consideration, it is indeed unreasonable to index Li_7PS_6 for comparison even if they have the same structure. According to your suggestion, we have replaced the XRD pattern of Li_7PS_6 with that of $\text{Li}_6\text{PS}_5\text{Cl}$ (LPSCl). The description in the manuscript (paragraph 2, page 4) and the revised Figure S1 in the supplementary information is shown as follow:

“The purity of synthesized LPSCl was determined by XRD patterns (Figure S1 in the supplementary information). The diffraction peaks are well indexed to standard LPSCl.”

Figure S1. XRD patterns of the synthesized LPSCl and standard LPSCl.

#Q1.2: Although the reported room-temperature ionic conductivity of the chloride argyrodite is within previously reported values (see e.g. Ohno et al. ACS Energy Lett, 2020, 5, 9, 910), the electrical equivalent circuit shown in Figure S2 is invalid for analyzing such a spectrum and is unlikely to have given such a value. Based on the inset shown in Figure S2, extrapolating the intercept at the real impedance axis (it should be ca. 70-75 Ohms), and then using the pellet dimensions to determine the ionic conductivity should be sufficient and more correct.

#A1.2: Thanks for your suggestion. As you mentioned, the electrical equivalent circuit shown in Figure S3 (the figure number has changed after revision) is invalid for analyzing such a spectrum after careful consideration. Therefore, we recalculated the ionic conductivity based on the extrapolated intercept at the real impedance axis and the pellet dimensions according to your suggestion. The thickness (H) and area (S) of the pellet are 765 μm and 0.196 cm^2 , respectively. The extrapolated intercept (R) is 65.5 Ω calculated from the inset in Figure S3. Based on the formula $\sigma = H/(S \times R)$, the ionic conductivity of LPSCl is $5.96 \times 10^{-3} \text{ S cm}^{-1}$. The revised description in paragraph 2, page 4 and Figure S3 in the supplementary information are displayed as follow:

“The electrolyte LPSCl has a high ion conductivity of $5.96 \times 10^{-3} \text{ S cm}^{-1}$ at room temperature (25 °C) as measured by electrochemical impedance spectroscopy (Figure S3).”

Figure S3. Nyquist plot of the synthesized LPSCl solid electrolyte at 25 °C.

#Q1.3: In Figure 3, the authors show that there are In-rich regions (close to the initial anode-solid electrolyte interface) and (regular?) Li-In dendrites that have grown up to the opposite cathode-solid electrolyte interface. It is however unclear to me how were these In-rich regions identified, and more importantly how do these compositional changes in the alloy affect the cell potential. Also what exactly is the mechanism for these spatially-resolved compositional changes in the alloy dendrites? Is the diffusivity of Li hindered due to the formation of vacancies as was also demonstrated by the Janek group with the LiMg alloy in contact with garnet electrolytes? (see: Krauskopf et al. Adv. Energy Mater. 2019, 9, 1902568). Since the authors are talking about growth mechanisms, there could/should be more describing the fundamentals of this process.

#A1.3: Thanks for your comments. Firstly, please allow me to point that the Li-In dendrite in In-rich region does not mean it has higher indium content or lower lithium content than “regular” Li-In dendrite. It means Li-In alloy occupies a higher proportion in the mixture of Li-In alloy and electrolyte in In-rich layer. The Li-In dendrites is classified based on the **morphology of dendrites at different regions**. The Li-In dendrites in In-rich layer exhibit a form of network enclosing the broken electrolyte particles, which is different from the morphologies in top and middle regions. However, Our description is indeed misleading, so we have removed the description “In-rich layer” in the manuscript and the revised discussion in paragraph 1, page 9 is described as follow:

“Figure 3g shows the morphology of Li-In dendrites **in the top region** (blue boxed area in Figure 3a). Different from the streak pattern in the middle region of Li-In dendrites, it displays a flame shape, which is similar as the morphology of the cell with 100 cycles (Figure 2c). From the perspective of time and space, it can be inferred that flame shape is the initial morphology of Li-In dendrites. **In the bottom region of Li-In dendrites**, the electrolyte is broken into smaller particles with less than 4 μm diameter as seen from Figure 3h (green boxed area). The particle diameter becomes smaller with getting closer to the bottom. Li-In dendrites exhibit a form of network

enclosing the broken electrolyte particles. Figure 3i shows the SEM image of transition region between the two morphologies of Li-In dendrites (black boxed area). The upper part exhibits streak pattern, which is consistent with the morphology in the middle region of Li-In dendrites. The lower part shows network formation, which is in accord with the bottom layer. Besides, the striped dendrites shown in Figure 3i is much denser than that in Figure 3b and Li-In alloy occupies a higher proportion in the mixture of Li-In alloy and electrolyte. Therefore, the damage for electrolyte caused by dendrites is aggravated as the cycling process goes on.”

The In-rich dendrite and “regular” Li-In dendrite were defined based on the dendrite morphology. As seen from the SEM images of Figure 3b and 3h, it can be clearly found that there exist significant morphological differences between In-rich dendrite and “regular” Li-In dendrites. The Li-In dendrites in In-rich layer (Figure 3h) exhibits a form of network enclosing the broken electrolyte particles, while the “regular” Li-In dendrites grow densely and laterally in stripes. Moreover, there exist clear boundaries between the two regions as shown in Figure 3i, by which In-rich region can be identified.

For the cell potential, as shown in Figure 1, no significant capacity fading or abnormal voltage change were observed during the ultra-long cycling (897 cycles) except for the last few cycles. Therefore, it can be concluded that the growth of Li-In dendrites will not affect the cell potential until it has a contact with cathode.

For the spatially-resolved compositional changes in the alloy dendrites as you mentioned, we guess that our unreasonable statement misleads you to think that there exists a spatial variation of composition. Honestly, we think lithium has a good diffusivity in indium matrix and the growth of Li-In dendrites will not hinder the diffusivity of lithium. As shown in Figure 3b, no significant cracks or vacancies are observed even if the Li-In dendrites grows vigorously in the solid electrolyte. On the contrary, it has a lower porosity than the compacted electrolyte because the expanded indium matrix induced by high current fills the pores and grain boundaries between electrolyte particles. Therefore, the composition of Li-In dendrites is spatially

uniform.

#Q1.4: In Figure 4, it is unclear if the dendrites imaged correspond to the In-rich phase or the “regular” Li-In dendrites. Also, is the composition of the interphase layer dependent on whether it is an In-rich dendrite or a “regular” Li-In dendrite?

#A1.4: Thanks for your comments. The Li-In dendrites shown in Figure 4 correspond to “regular” Li-In dendrites. We are sorry for our unclear expression. We have added the corresponding statement in paragraph 2, page 12, which is described as:

“Figure 4a and 4b show the STEM-HAADF images of Li-In dendrites **in the middle region** at low and high magnifications, respectively.”

The composition of the interphase layer does not depend on whether it is an alloy-rich dendrite or a “regular” Li-In dendrite. The interphase layer is formed by the slight interface reaction between metal indium and electrolyte LPSCl when contacting. As demonstrated by STEM-EDX analysis, STEM-EELS analysis and AIMD calculation, the composition of the interphase layer is mainly InS. As we have illustrated in Q1.3, the composition of Li-In dendrites is spatially uniform. Therefore, the composition of interphase layer is independent with spatial region. More importantly, both alloy-rich dendrite and “regular” Li-In dendrite have the same essence and are just at different development stages. As the cycle progresses, “regular” Li-In dendrites continue to grow towards the opposite side, and alloy-rich dendrites will become “regular” Li-In dendrites.

#Q1.5: I am not a theoretical expert, so the following comments should be taken with that statement in mind:

- a. Why is the simulation done only considering pure indium as the anode and not an Li-In alloy?
- b. The authors mention that they used the crystalline compounds InS and InCl to calculate the RDFs, however, I am not convinced that these compounds actually

exist, especially the indium sulfide (In^{1+} can be stable and makes InCl more reasonable). Can the authors comment a bit on what structures were actually considered for this analysis?

c. What exactly happens to the Li and P atoms once In-S and In-Cl bonds start forming?

#A1.5: Thanks for your comments. For the first question, we chose pure indium for simulation for two reasons. On one hand, the interface reaction between electrolyte LPSCl and Li contained in Li-In alloy can be ignored. The STEM-HAADF images shown in Figure 4a-4c and excellent cycling performance shown in Figure 1a exclude the formation of Li_2S and LiCl at the interface. It is well known that the difference in atomic number at different regions of sample can be reflected from the STEM image contrast. The sample with higher atomic number has higher brightness in high-angle annular dark field (HAADF) image. Figure 4b and 4c show the locally enlarged STEM-HAADF images of Li-In dendrites, in which Li-In dendrites, sulfide electrolyte and the formed interphase layer can be clearly distinguished. Calculated from the mass ratio of lithium to indium in Li-In alloy, the molar ratio of lithium to indium is 1:3. The average atomic number of Li-In alloy is 37.5, while the average atomic number of electrolyte LPSCl is 10. Therefore, the Li-In dendrite is much brighter than the electrolyte in the HAADF image, which is in accordance with Figure 4a-4c. The brightness of the interphase layer is between the Li-In alloy and electrolyte, thus the average atomic number of the formed product is in the range of 10~37.5, excluding the generation of Li_2S (7.3) and LiCl (10).

Besides, if LPSCl and Li react at the interface, the formed Li_2S will lead to an increase in interface impedance¹⁻³, further causing capacity fading of the cell. However, as shown in Figure 1a, no significant capacity fading is observed during the long-term cycling. Therefore, the interface reaction between Li contained in Li-In alloy and LPSCl can be ignored.

On the other hand, there definitely exists interface reaction between metal In and electrolyte LPSCl. As demonstrated by the STEM-EDX analysis (Figure 4d), the

main elements of the interphase layer are indium and sulfur. STEM-EELS result further demonstrates the generation of new phase (Figure 5b). Therefore, we performed simulation only considering pure indium. In order to make the description more clearly, we have added the corresponding explanations in the manuscript.

The description in paragraph 2, page 12 has been revised as follow:

“Figure 4a and 4b show the STEM-HAADF images of Li-In dendrites in the middle region at low and high magnifications, respectively. It is clearly illustrated that there exists a 15 nm-thick interphase layer at the LPSCI-LiIn interface, which may be composed of compounds such as Li_2S , LiCl , InS , InCl , etc. It is well known that the STEM image contrast can reflect the difference in atomic number at different regions of sample. The sample with higher atomic number has higher brightness in high-angle annular dark field (HAADF) image. The average atomic number of Li-In alloy is 37.5, while that of electrolyte LPSCI is 10. Therefore, the Li-In dendrite is much brighter than the electrolyte in the HAADF image, which is in accordance with Figure 4a-c. The brightness of the interphase layer is between the Li-In alloy and electrolyte, which excludes the generation of Li_2S (7.3) and LiCl (10). Besides, the excellent cycling performance shown in Figure 1a further excludes the formation of Li_2S , otherwise the capacity will decrease due to the increased interface impedance⁶⁻⁸. Therefore, the interface reaction between Li contained in Li-In alloy and LPSCI can be ignored.”

The description in paragraph 1, page 13 has been revised as follow:

“The interphase layer is mainly composed of indium and sulfur elements. Some indium-sulfur compounds may be generated at the interphase layer. Therefore, the growth of Li-In dendrites is probably caused by the interaction between metal In and electrolyte LPSCI. Although metal Li is not directly involved in the interface reaction, it may act as a catalyst and facilitate this process. The role of metal Li in the growth of Li-In dendrites deserves further investigation.”

The description in paragraph 1, page 16 has been revised as follow:

“Then, first-principles calculations were further performed to investigate the

interface reaction between metal In and LPSCl electrolyte. As mentioned above, the interface reaction between metal Li and electrolyte LPSCl has been excluded. The interface reaction mainly occurs between metal In and electrolyte LPSCl. Therefore, only pure indium was used for simulation.”

1. Wenzel S. et al. Direct observation of the interfacial instability of the fast ionic conductor $\text{Li}_{10}\text{GeP}_2\text{S}_{12}$ at the lithium metal anode. *Chem. Mater.* **28**, 2400-2407 (2016).
2. Wenzel S, et al. Interphase formation and degradation of charge transfer kinetics between a lithium metal anode and highly crystalline $\text{Li}_7\text{P}_3\text{S}_{11}$ solid electrolyte. *Solid State Ionics* **286**, 24-33 (2016).
3. Wenzel S, et al. Interfacial reactivity and interphase growth of argyrodite solid electrolytes at lithium metal electrodes. *Solid State Ionics* **318**, 102-112 (2018).

For the second question, please allow me to point that the description in the question “they used the crystalline compounds InS and InCl to calculate the RDFs” is not accurate enough. The RDFs were calculated based on the results of AIMD. In order to determine the product, we searched the Materials Project database for stable compounds that may be generated by In, S and Cl elements. Among these possible compounds, only the RDFs of InS and InCl match well the simulated results. Therefore, we think the generated compounds are InS and InCl. The structural stability can be evaluated by energy above hull (E_{hull}). Material with an $E_{hull} < 20$ meV/atom can be considered as a relatively stable phase⁴. The E_{hull} of **InS (mp-19795)** and **InCl (mp-23276)** are both 0 meV/atom, thus they are ultra-stable materials. The E_{hull} of InS (mp-630528) is 28 meV/atom that may decomposes into InS (mp-19795). Therefore, InS (mp-19795) and InCl (mp-23276) are the final products at the LPSCl-In interface. In order to make the description more clearly, we have revised the discussion in paragraph 2, page 16 as follow:

“The upper part in Figure 5d shows the RDF evolutions of In-S, In-Cl and P-S pairs during the simulation. The lower part plots the RDFs of In-S, In-Cl and P-S before MD and possible products (InS and InCl) provided as reference materials. The structures of known crystalline compounds were obtained from Materials Project (MP)

database. Comparing the RDFs evolution of In-S and In-Cl during the simulation, new peaks of In-S (~ 2.8 Å) and In-Cl (~ 2.9 and 3.5 Å) are generated, which means the formation of In-S and In-Cl bonds. The initial formation of In-S and In-Cl bonds is at ~ 1 ps. The intensity of In-S and In-Cl bonds gradually increases and then keep stable during the simulation. The decrease of peak intensity for P-S RDF indicates the break of the P-S bonds.

In order to determine the product, we searched MP database for stable compounds that may be generated by In, S and Cl elements. Among these possible compounds, only InS and InCl match well the simulated results and have high structural stability. The structural stability can be evaluated by energy above hull (E_{hull}). Material with an $E_{hull} < 20$ meV/atom can be considered as a relatively stable phase. The E_{hull} of InS (mp-19795) and InCl (mp-23276) are both 0 meV/atom, thus they are ultra-stable materials. The E_{hull} of InS (mp-630528) is 28 meV/atom that may decomposes into InS (mp-19795). Therefore, InS (mp-19795) and InCl (mp-23276) are the final products at the LPSCl-In interface. However, due to the low content of Cl in the LPSCl electrolyte, the amount of InCl is much less than that of InS, which can be reflected from the number of In-S bonds and In-Cl bonds in Figure 5c. The average atomic number of InS (28.5) is also in the range of 10~37.5. Therefore, InS should be the main reaction product.”

4. Tang H. M. et al. Probing solid solid interfacial reactions in all-solid-state sodium-ion batteries with first-principles calculations. *Chem. Mater.* **30**, 163-173 (2018).

For the third question, the changes of Li and P atoms can be reflected from the RDF evolutions of Li-P, Li-S and Li-Cl pairs, which is shown as follow. The red lines represent the RDFs of standard materials with $E_{hull} \leq 20$ meV/atom. During the simulation, some new peaks appear in RDF of Li-P in the range of 2.5 Å~ 3.3 Å, indicating the formation of Li-P bond and generation of Li_xP_y . Similarly, the new peak at ~ 2.5 Å in Li-S RDF and new peaks in the range of 2.2 - 2.7 Å in Li-Cl RDF indicate the generation of Li_2S and LiCl , respectively. Moreover, the peak in Li-Cl RDF is consistent with that of LiCl RDF (mp-1185319, $E_{hull} = 20$ meV/atom) in the first 8 ps,

while it agrees well with that of LiCl RDF (mp-22905, $E_{hull} = 0$ meV/atom) in the last 2 ps, implying a LiCl conversion to a more stable structure. Besides, the brightness of pristine Li-P, Li-S and Li-Cl at the initial state gradually becomes weak with time, which means the decomposition of electrolyte LPSCl. This section has been added into the supplementary information. The description in the manuscript (paragraph 1, page 17) has been revised as:

“Besides, the changes of Li-P, Li-S and Li-Cl bonds for electrolyte LPSCl during the simulation are clearly illustrated in Figure S12.”

Figure S12. Evolutions of the radial distribution functions (RDFs) vs. AIMD simulation time at room temperature (300 K), where RDFs of Li-P, Li-S and Li-Cl are at LPSCl-In interface.

Thank you again for your suggestions.

For the second reviewer:

#Q2.1: Information on the Li-In electrode is very vague. It says the total weight is approx. 25 g and the Li:In ratio 2:98. But what are the relevant values? Electrode thickness? Capacity of Li of this electrode is not mentioned at all. Is it also 4 mAh/cm²? How was such little Li applied?

#A2.1: Thanks for your comments. First of all, we mistakenly wrote 25 mg as 25 g in the manuscript. We are very sorry for the misunderstanding caused by our mistake. We have revised it in paragraph 2, page 23 as follow:

“Next, Li-In alloy (~ 25 mg) was placed on the other side of the electrolyte layer

with a SS foil as collector.”

The thickness of the Li-In anode can be obtained from the SEM image shown in Figure S5. It is around 40 μm . According to the weight ratio of Li and In, the metal lithium and metal indium is 0.5 mg and 24.5 mg, respectively. The capacity of lithium is 1.9 mAh cm^{-2} based on the theoretical specific capacity of 3860 mAh g^{-1} . The capacity of metal indium is 4.8 mAh cm^{-2} calculated from the theoretical specific capacity of 194 mAh g^{-1} . Therefore, the capacity of anode is 6.7 mAh cm^{-2} and much higher than that of cathode. Therefore, the capacity of the cell is determined by the cathode loading. Moreover, it should be noted that the metal indium in Li-In anode is just used for storing lithium transporting from the cathode, rather than the source of lithium. The metal lithium contained in Li-In anode is used to supplement the lithium lost by the cathode during the cycling. Therefore, the capacity of the cell is limited by the cathode as long as the Li-In anode can accommodate the lithium ions transported from the cathode.

Moreover, we did not directly operate on such a little lithium. In the actual preparation, a large piece of lithium and a large piece of indium with a weight ratio of 2:98 were pressed together under a high pressure of 760 MPa. Due to the good deformation properties of lithium and indium, Li metal easily diffuses into the In matrix and the uniformity of Li-In anode can be reflected from Figure S5. Then the obtained Li-In alloy was cut into disks with diameter of 10 mm to assemble cells.

#Q2.2: Same thing for the Li electrode, which is apparently only 5 μm thick.

How was this done and confirmed?

#A2.2: Thanks for your comments. Firstly, please allow me to point that the thickness of Li anode used in our testing is $\sim 100 \mu\text{m}$, instead of 5 μm . The Li electrode shown in Figure 6 is only 5 μm thickness. This is because lithium anode was not completely stripped from the current collector during the disassembly process, which results in a significantly reduced thickness of Li anode attached to the electrolyte. In order to avoid misunderstanding, we have explained it in paragraph 2, page 19 and paragraph

2, page 23:

“Herein, it should be noted that the ultra-thin Li anode ($\sim 5 \mu\text{m}$) shown in Figure 6 is also induced by the cell disassembly. Lithium anode was not completely stripped from the current collector and adhered to the stainless steel. The thickness of Li anode used in the testing is $\sim 100 \mu\text{m}$.”

“For the reference cell LNO@NCM622//LPSCI//Li, the Li-In anode was replaced by Li metal (**$\sim 100 \mu\text{m}$ thickness**) and the weight of composite cathode is reduced to 10 mg. The pressure was changed from 760 MPa to 100 Mpa with other preparation conditions remaining consistent.”

#Q2.3: There is not a real proof for that the dendrites are really a Li-In alloy. Of course, this is not easy but since there is no evidence it is only a hypothesis. Could it be not simply formation of InS along the grain boundary as also suggested from theory? The authors could check this by simply making another cell and simply let it rest for long time (without cycling). Maybe the In reacts with the solid electrolyte?

#A2.3: Thanks for your comments. Firstly, lithium ions are transformed into metal lithium in the anode during charging. As the earliest part of anode in contact with electrolyte, dendrites firstly became the diffusion channels of lithium ions and the matrix for storing metal lithium. Therefore, it is unreasonable only having metal indium in the dendrites, but no metal lithium. In the most cases, it is in a state of Li-In alloy.

Secondly, if the growth of dendrites is simply induced by the formation of InS without the influence of cycling, there is no enough force to drive metal indium growing into the electrolyte. InS can only be formed at the anode-electrolyte interface, instead of electrolyte interior.

According to your suggestion, we reassembled a cell and let it rest for 60 days. The cross-sectional SEM images are shown as follow (Figure 2a). No significant Li-In dendrites are observed at anode-electrolyte interface. A small amount of Li-In anode

enters the electrolyte observed from the magnified SEM image of the red box area, which is caused by the high pressure (760 MPa) during cell assembly. Owing to the great deformation property of Li-In alloy, it easily fills the pores and voids at the bottom region of electrolyte layer. This part of Li-In anode improves the interface contact between Li-In anode and electrolyte and will not constantly expand to the opposite, which is favorable for ion transport. Moreover, the Li-In anode pressed into the electrolyte layer has a quite different morphology with Li-In dendrites, which can be clearly observed from Figure 2. Therefore, the growth of Li-In dendrites is not simply caused by the formation of InS at the interface.

In order to make the results more convincing, the SEM images of fresh cell without cycling in Figure 2a are replaced by the images of the cell that rests for 60 days. The revised Figure 2 is shown as follow. The description in the manuscript (paragraph 1, page 6) is revised as:

“Figure 2a shows the cross-sectional SEM image for the cell that rests for 60 days without cycling. Cathode LNO@NCM622, electrolyte LPSCl and Li-In anode can be clearly distinguished from the SEM images. **Owing to the great deformation property of Li-In alloy, it easily fills the pores and voids at the bottom region of electrolyte layer when the cell is assembled under a high pressure, which ensures the intimate interface contact.”**

Figure 2. Cross-sectional SEM images for the cells LNO@NCM622//LPSCI//Li-In with different cycling numbers. **a** The cell resting for 60 days without cycling. **b** The cell with 100 cycles. **c** The cell with 897 cycles.

#Q2.4: If it is true that Li-In dendrites grow, they should also do this in cells with liquid electrolyte. Did the authors consider doing the same experiment in cells with liquid electrolyte? This would support the claim that Li-In alloys can form dendrites.

#A2.4: Thanks for your comments. According to your suggestions, we have assembled cells with liquid electrolyte. 1 M LiPF₆ in 1:1 vol/vol EC/DEC with FEC additive was used as liquid electrolyte. The weight of cathode active material in liquid

cell is consistent with that in all-solid-state cell. The liquid cell was cycled 300 times at 3.8 mA cm^{-2} ($\sim 1\text{C}$) at $25 \text{ }^\circ\text{C}$. The cycling performance of the liquid cell NCM622//LiPF₆//Li-In is shown as follow. The poor capacity retention is induced by the poor interface compatibility between cathode and liquid electrolyte, which is not described in detail here.

Figure S8. Long-term cycling performance of the liquid cell NCM622//LiPF₆//Li-In at 3.8 mA cm^{-2} at $25 \text{ }^\circ\text{C}$.

Then the cell was disassembled and the Li-In anode was dried at $80 \text{ }^\circ\text{C}$ for 40 hours to remove the liquid electrolyte. The surface and cross-section SEM images of the cycled Li-In anode in liquid cell are shown as follow. It can be clearly observed that the surface of Li-In anode in Figure S9a is hillocky and has many cracks. The SEM image and EDX mapping of cross section in Figure S9b-S9d further demonstrate the morphology difference between surface and interior. The surface of Li-In anode is obviously loose and porous, while the interior is rather dense. This part of porous structure at the surface can be considered as Li-In dendrites in liquid cell, which is caused by the repeated expansion and contraction of Li-In anode during cycling. Observed from Figure S9b, Li-In dendrites has around $57 \text{ }\mu\text{m}$ thickness after 300 cycles.

However, the morphology of Li-In dendrites in liquid cell is different from that in solid cell, which can be attributed to the difference of pressure applied on Li-In anode.

For the cell with solid electrolyte, it runs under a high pressure to maintain an intimate interface contact. When the Li-In anode expands during charging, there is not enough space to accommodate the expanded Li-In anode. Therefore, it can only grow into the electrolyte along the pores and grain boundaries. However, for the cell with liquid electrolyte, there is almost no pressure applied on Li-In anode and it can expand freely into the liquid electrolyte. The Li-In dendrites in liquid cell are less likely to induce short circuit due to the obstruction of separator and smoothness of dendrites. But the repeated expansion and contraction during cycling will cause surface cracking and bulging, which have been demonstrated by Figure S9a and S9b.

Figure S9. SEM images for the Li-In anode cycled 300 times in liquid cell. **a** SEM image for the surface of Li-In anode. **b** SEM image for the cross section of Li-In anode. **c** Magnified SEM image of Li-In dendrites in red boxed area. **d** Corresponding EDX mapping of element In in red boxed area.

In order to prove the universality of Li-In dendrite growth, solid cells with other typical sulfide electrolytes $\text{Li}_{10}\text{GeP}_2\text{S}_{11}$ (LGPS) and $\text{Li}_7\text{P}_3\text{S}_{11}$ (LPS) were also tested. The testing conditions are the same as that of the cell with LPSCl electrolyte in the manuscript. The long-term cycling performance of the cells with LPS and LGPS is

shown as follow. Due to the higher ion conductivity of LGPS, it has a higher discharge capacity than LPS in the early stage of cycling. However, the cell with LPS has higher capacity retention owing to the better electrochemical stability.

Figure S10. Long-term cycling performance of the cells NCM622//LGPS//Li-In and NCM622//LPS//Li-In at 3.8 mA cm⁻² at 25 °C.

The cells after 300 cycles were stopped and the cross-sectional SEM images of Li-In dendrites in LGPS and LPS are shown as follow. It can be clearly found that Li-In dendrites exist in both LGPS and LPS electrolytes and exhibit stripe pattern, which is consistent with the morphology in LPSCI electrolyte. However, the stripes in LGPS is much denser than that in LPS and the growth rate of Li-In dendrites in LGPS is higher than that in LPS or LPSCI, which is attributed to the higher reactivity of LGPS with Li-In alloy. Therefore, improving the electrochemical stability of electrolyte, reducing the porosity of electrolyte and introducing some other elements in the alloy are effective means to inhibit Li-In dendrites.

Figure S11. Cross-sectional SEM images of the electrolyte-anode interface after 300 cycles. **a** Cross-sectional SEM image and **b** magnified SEM image of Li-In dendrites in LGPS electrolyte. EDX mapping of **c** In and **d** Ge in the red boxed area. **e** SEM image of Li-In dendrites in LPS electrolyte. **f** Corresponding EDX mapping of In.

According to your suggestions, this part has been added in the manuscript and supplementary information. The added discussion in the manuscript (paragraph 3, page 11) has been described as follow:

“Similarly, Li-In dendrites were also observed in liquid cell and solid cells with other typical sulfide electrolytes $\text{Li}_{10}\text{GeP}_2\text{S}_{11}$ (LGPS) and $\text{Li}_7\text{P}_3\text{S}_{11}$ (LPS). The cathode loading and testing current are the same as that of the cell with LPSCl electrolyte. The long-term cycling performance of the liquid cell and SEM images of

dried Li-In anode are shown in Figure S8 and Figure S9. The long-term cycling performance of the cell with LGPS/LPS and SEM images of Li-In anode with 300 cycles are displayed in Figure S10 and Figure S11. The detailed analysis are clarified in the supplementary information. It can be clearly observed that Li-In dendrites exist in LGPS, LPS and liquid electrolyte. Moreover, the growth of Li-In dendrites will be accelerated in high-reactivity sulfide electrolytes. Therefore, improving the electrochemical stability of electrolyte, reducing the porosity of electrolyte and introducing some other elements in the alloy are effective means to inhibit Li-In dendrites. The Li-In dendrites of other types of electrolytes are not discussed here in detail.”

The added description in the conclusion (paragraph 1, page 22) is revised as follow:

“The growth of Li-In dendrites can be inhibited by improving the electrochemical stability of electrolyte or Li-In anode and reducing the porosity of electrolyte.”

#Q2.5: Studies on dendrites are highly erratic, so one cell may fail after 800 cycles, the other one at 100 cycles. It is clear that long-time measurements are time consuming but the authors should at least comment on how many experiments with cell failure were done.

#A2.5: Thanks for your comments. The cell cycled 100 times in Figure 2b did not fail. We manually stopped it at 100th cycle to observe the growth morphology of Li-In dendrites at a certain stage of cycling. In order to avoid misunderstanding, we have added the explanation in paragraph 1, page 7:

“For the cell cycled 100 times (**without short circuit**) shown in Figure 2b, different from the fresh state, Li-In alloy grows into the electrolyte layer for around 30 μm and exhibits flame shape at the anode-electrolyte interface. For the dead cell with 897 cycles (**short circuit**), the Li-In alloy exhibits a striking growth towards

electrolyte interior with around 500 μm , nearly having a contact with cathode.”

We have ever performed 9 sets of repeated experiments at the same testing conditions for the cell LNO@NCM622//LPSCI//Li-In. The results are shown as follow (Figure S6). It can be clearly found that all the cells are short-circuited after a long cycling, which can be reflected from the sharp decrease of Coulomb efficiency in Figure S6a. Therefore, the growth of Li-In dendrites in sulfide electrolyte is universal when the cell is cycled at high loading and high current. The distribution of cycling life for the nine cells is shown in Figure S6b. Obviously, the cell life mainly distributes in the range of 800~1000 cycles. Therefore, the cell with 897 cycles was selected as a typical representative in the manuscript. We have added the corresponding description in the manuscript (paragraph 1, page 5), which is revised as follow:

“Nine sets of repeated experiments with the same testing conditions have been performed for the cell LNO@NCM622//LPSCI//Li-In for further verification. The long-cycling performance and distribution of cycling life are shown in Figure S6. All the cells have a short circuit after a long cycling with cell life mainly distributed in the range of 800~1000 cycles, which demonstrates the universality of cell failure in ASSLBs with Li-In anode.”

Figure S6. Nine sets of repeated experiments for the cells LNO@NCM622//LPSCI//Li-In. **a** Long cycling performance at 3.8 mA cm⁻² at 25 °C. **b** Distribution of cycling life for the nine cells.

#Q2.6: Authors claim that the dendrites grow only along grain boundaries but

there is no information on whether the solid electrolyte is really dense. SEM is not sufficient here. Did the authors try to determine the density of the pellet? Does it correspond to the theoretical density? Otherwise the dendrites might grow along the pores.

#A2.6: Thanks for your suggestions. After careful consideration, we think the description “the Li-In dendrites grow along the grain boundaries” in the manuscript is not accurate enough. As you mentioned, there is no information on whether the solid electrolyte is really dense. Actually, sulfide solid electrolyte is not dense. It has a relative density of 0.85 or about 15% porous even it has been pressed under a high pressure^{5,6}. As described in the manuscript, the growth of Li-In dendrites is caused by the expansion of the indium matrix when the cell is cycled at high loading and high current, like liquid filling the solid electrolyte. During the expansion process, both grain boundaries and pores will be filled. Therefore, Li-In dendrites not only grow along the grain boundaries, but also along the pores. However, compared with Li dendrites, the expansion of indium matrix has much lower growth stress, without significant structure damage on electrolyte layer. According to your suggestion, the corresponding descriptions in the manuscript have been revised.

The description in paragraph 2, page 17 has been revised as follow:

“Obviously, **grain boundaries and pores** are the preferential expansion channels due to the least resistance.”

The description in paragraph 1, page 18 has been revised as follow:

“Therefore, the formed Li-In dendrites fill the **grain boundaries and pores** like liquid and tightly enclose the electrolyte particles, which enables the structural stability of the electrolyte and even improves the overall density.”

The description in paragraph 2, page 19 has been revised as follow:

“Li dendrites not only grow along the **grain boundaries and pores**, but also destroy the electrolyte structure and induces much cracks, which results in a rapid short circuit within several cycles.”

The description in paragraph 1, page 20 has been revised as follow:

“This is because the growth of Li dendrites is induced by the non-uniform Li deposition that prefers to form whiskers, while Li-In dendrites is caused by the volume expansion and slight interface reaction that prefers to fill the **grain boundaries and pores.**”

5. Whiteley J. M. et al. Ultra-thin solid-state Li-ion electrolyte membrane facilitated by a self-healing polymer matrix. *Adv. Mater.* **27**, 6922-6927 (2015).

6. Wang S. et al. High-conductivity free-standing Li₆PS₅Cl/poly(vinylidene difluoride) composite solid electrolyte membranes for lithium-ion batteries. *J. Mater.* **6**, 70-76 (2020).

#Q2.7: Figure 2 shows already lot of cracks/pores before cycling. Is it then reasonable to have the discussion on crack formation in case of lithium cells and not crack formation for the Li-In cells?

#A2.7: Thanks for your comment. As you mentioned, there indeed exist many cracks and pores before cycling, which is determined by the nature of sulfide electrolyte. As we have mentioned in Q2.6, the porosity of sulfide solid electrolyte is about 15%. Moreover, due to the brittle nature of sulfide electrolytes, it is inevitable to produce microcracks during the pressing process⁵⁻¹¹, which is consistent with our experimental result shown in Figure 2a. Therefore, the discussion on crack formation for Li/Li-In cells is indeed unreasonable. A more accurate description is that the growth of Li dendrites and Li-In dendrites have quite different effects on **crack propagation**. Comparing the electrolyte with Li-In dendrites (Figure 3b) and with Li dendrites (Figure 6b) shown as follow, there exist significant structural differences. Li-In dendrites fills the grain boundaries and pores during the expansion, contributing to a highly dense electrolyte structure. Conversely, Li dendrites have much higher growth stress due to the difference in growth mechanism, resulting in a severe crack propagation and a loose electrolyte structure. According to your suggestion, the corresponding descriptions in the manuscript have been revised.

The description in paragraph 2, page 19 has been revised as follow:

“Li dendrites not only grow along the grain boundaries and pores, but also destroy

the electrolyte structure and induces **severe crack propagation**, which results in a rapid short circuit within several cycles.”

The description in paragraph 1, page 20 has been revised as follow:

“The growth of Li dendrites causes **severe crack propagation** due to the stress concentration and high reactivity, leading to a loose and porous electrolyte structure.”

Figure R1. Difference of dendrite morphology with Li-In anode and Li anode.

5. Whiteley J. M. et al. Ultra-thin solid-state Li-ion electrolyte membrane facilitated by a self-healing polymer matrix. *Adv. Mater.* **27**, 6922-6927 (2015).
6. Wang S. et al. High-conductivity free-standing $\text{Li}_6\text{PS}_5\text{Cl}$ /poly(vinylidene difluoride) composite solid electrolyte membranes for lithium-ion batteries. *J. Mater.* **6**, 70-76 (2020).
7. Inada T. et al. All solid-state sheet battery using lithium inorganic solid electrolyte, thio-LISICON. *J. Power Sources* **194**, 1085-1088 (2009).
8. Nam Y. J. et al. Bendable and thin sulfide solid electrolyte film: a new electrolyte opportunity for free-standing and stackable high-energy all-solid-state lithium-ion batteries. *Nano Lett.* **15**, 3317-3323 (2015).
9. Yersak T. et al. Hot pressed, fiber-reinforced $(\text{Li}_2\text{S})_{70}(\text{P}_2\text{S}_5)_{30}$ solid-state electrolyte separators for Li metal batteries. *ACS Appl. Energy Mater.* **2**, 3523-3531 (2019).
10. Zhang Y. B. et al. Free-standing sulfide/polymer composite solid electrolyte membranes with high conductance for all-solid-state lithium batteries. *Energy Stor. Mater.* **25**, 145-153 (2020).
11. Luo S. T. et al. A high energy and power all-solid-state lithium battery enabled by modified sulfide electrolyte film. *J Power Sources* **485**, 229325 (2021).

#Q2.8: Authors state that they remove the electrolyte and the cathode material using deionized water by a chemical reaction. It is impossible to dissolve cathode material by water. It is hard to understand what the authors.

#A2.8: Thanks for your comments. The cathode composite was prepared using active material LNO@NCM622 and solid electrolyte LPSCl with a mass ratio of 7:3. When the cathode is soaked in deionized water, the sulfide electrolyte contained in the cathode is removed by chemical reaction. The remaining active material cannot maintain a complete structure and can be washed away by deionized water. In order to make the description more accurately, the description in paragraph 2, page 10 has been revised as follow:

“In order to more clearly observe the morphology of Li-In dendrites without the influence of electrolyte, a part of the dismantled specimen of the LNO@NCM622//LPSCl//Li-In cell was soaked and washed in deionized water for 2 hours and then dried for 24 hours. **The LPSCl electrolyte can be removed via chemical reaction. The remaining active material NCM622 cannot maintain a complete structure and can be washed away by deionized water.**”

#Q2.9: Interface: Authors state that InS forms, but what about Li₂S?

#A2.9: Thanks for your comment. It is well known that the difference in atomic number at different regions of sample can be reflected from the STEM image contrast. The sample with higher atomic number has higher brightness in high-angle annular dark field (HAADF) image. Figure 4b and 4c show the locally enlarged STEM-HAADF images of Li-In dendrites, in which Li-In dendrites, sulfide electrolyte and the formed interphase layer can be clearly distinguished. Calculated from the mass ratio of lithium to indium in Li-In alloy, the molar ratio of lithium to indium is 1:3. The average atomic number of Li-In alloy is 37.5, while the average atomic number of electrolyte LPSCl is 10. Therefore, the Li-In dendrite is much brighter than the electrolyte in the STEM-HAADF image, which is in accordance with Figure 4a-4c. The atomic number of InS is 28.5, thus its brightness is between

the Li-In alloy and electrolyte, which is consistent with the brightness of interphase shown in Figure 4b and 4c. However, the atomic number of Li_2S is 7.3, which is darker than the electrolyte. If Li_2S forms at the interface, the interphase layer should be the darkest, which is apparently incompatible with the experiments.

Besides, the formation of Li_2S will lead to an increase in interface impedance¹⁻³, further causing capacity fading of the cell. However, as shown in Figure 1a, no significant capacity fading is observed during the long-term cycling.

Therefore, no significant Li_2S generates at the interface as demonstrated from the contrast of STEM image and excellent capacity retention. According to your comment, we have added the explanation for STEM-HAADF image and cycling performance in the manuscript.

The description in paragraph 2, page 12 has been revised as follow:

“Figure 4a and 4b show the STEM-HAADF images of Li-In dendrites in the middle region at low and high magnifications, respectively. It is clearly illustrated that there exists a 15 nm-thick interphase layer at the LPSCI-LiIn interface, which may be composed of compounds such as Li_2S , LiCl , InS , InCl , etc. It is well known that the STEM image contrast can reflect the difference in atomic number at different regions of sample. The sample with higher atomic number has higher brightness in high-angle annular dark field (HAADF) image. The average atomic number of Li-In alloy is 37.5, while that of electrolyte LPSCI is 10. Therefore, the Li-In dendrite is much brighter than the electrolyte in the HAADF image, which is in accordance with Figure 4a-4c. The brightness of the interphase layer is between the Li-In alloy and electrolyte, which excludes the generation of Li_2S (7.3) and LiCl (10). Besides, the excellent cycling performance shown in Figure 1a further excludes the formation of Li_2S , otherwise the capacity will decrease due to the increased interface impedance⁶⁻⁸. Therefore, the interface reaction between Li contained in Li-In alloy and LPSCI can be ignored.”

The description in paragraph 1, page 13 has been revised as follow:

“The interphase layer is mainly composed of indium and sulfur elements. Some

indium-sulfur compounds may be generated at the interphase layer. Therefore, the growth of Li-In dendrites is probably caused by the interaction between metal In and electrolyte LPSCl. Although metal Li is not directly involved in the interface reaction, it may act as a catalyst and facilitate this process. The role of metal Li in the growth of Li-In dendrites deserves further investigation.”

The description in paragraph 1, page 17 has been revised as follow:

“However, due to the low content of Cl in the LPSCl electrolyte, the amount of InCl is much less than that of InS, which can be reflected from the number of In-S bonds and In-Cl bonds in Figure 5c. The average atomic number of InS (28.5) is also in the range of 10~37.5. Therefore, InS should be the main reaction product.”

1. Wenzel S. et al. Direct observation of the interfacial instability of the fast ionic conductor $\text{Li}_{10}\text{GeP}_2\text{S}_{12}$ at the lithium metal anode. *Chem. Mater.* **28**, 2400-2407 (2016).
2. Wenzel S, et al. Interphase formation and degradation of charge transfer kinetics between a lithium metal anode and highly crystalline $\text{Li}_7\text{P}_3\text{S}_{11}$ solid electrolyte. *Solid State Ionics* **286**, 24-33 (2016).
3. Wenzel S, et al. Interfacial reactivity and interphase growth of argyrodite solid electrolytes at lithium metal electrodes. *Solid State Ionics* **318**, 102-112 (2018).

#Q2.10: Page 10: Statement on high/low current. The paper only shows results at one current density. I can not follow the discussion at this point.

#A2.10: Thanks for your comment. We have ever performed the experiment at 2 mA cm^{-2} ($\sim 0.5\text{C}$) with 900 cycles as a contrast. The cycling performance is shown as follow (Figure S13). The cell achieves an excellent capacity retention of 87% with 900 cycles. Then the cell was disassembled for SEM analysis. Observed from the cross-sectional SEM images shown as follow (Figure S14), Li-In dendrites still exist at 2 mA cm^{-2} but the growth rate is much slower than 3.8 mA cm^{-2} . The thickness of Li-In dendrites at 2 mA cm^{-2} is $\sim 206 \mu\text{m}$, while it is $\sim 457 \mu\text{m}$ at 3.8 mA cm^{-2} with the same cycling number. The absence of Li-In anode in the bottom left corner in Figure S14a is caused by the disassembly process. Therefore, the growth rate of Li-In

dendrites is positively associated with cycling current.

According to your suggestion, we have added these results and discussion in supplementary information. The description in the manuscript (paragraph 2, page 17) is revised as:

“The experimental results shown in Figure S13 and Figure S14 demonstrates that the growth rate of Li-In dendrites is positively associated with cycling current.”

Figure S13. Long-term cycling performance of NCM//LPSC1//Li-In at 2 mA cm⁻² at 25 °C.

Figure S14. Cross-sectional SEM images of the cell **a** at 2 mA cm^{-2} and **b** at 3.8 mA cm^{-2} , respectively. Magnified SEM images of **c** the red boxed area and **d** the yellow boxed area.

#Q2.11: I find the title quite misleading as it does not highlight the major finding of the paper. Besides, “High energy and Power” is not addressed in this paper, so why mentioning this? With a 25 g In electrode and such a very thick solid electrolyte the cell will have very low energy and power. Did the authors calculate this?

#A2.11: Thanks for comments. We mistakenly wrote “25 mg” as “25 g” in the manuscript. We apologize again for our carelessness. Energy (Wh kg^{-1}) and power (W kg^{-1}) are positively correlated with the cathode loading and current density, respectively. “High energy” in the title corresponds to the high loading of the cell (4 mAh cm^{-2}) and “high power” corresponds to the high current (3.8 mA cm^{-2}). Importantly, high loading and high current are the necessary conditions for the growth

of Li-In dendrites. Based on the experimental results and theoretical calculation, we think the growth of Li-In dendrites is induced by the expansion of indium matrix when a large amounts of lithium ions enters into the indium during charging. The high current increases the expansion speed, and the high loading extends the single expansion time of the indium matrix. Therefore, the growth rate of Li-In dendrites in high-energy and high-power cells will be significantly accelerated. However, as you mentioned, we highlighted high energy and power in the title, but did not address it in the paper. Therefore, we have added the corresponding description in paragraph 1, page 18, which is revised as follow:

“High loading and high current are the necessary conditions for the growth of Li-In dendrites. Therefore, it is more likely to occur in high-energy and high-power batteries, which should be given attention in the testing and application of ASSLBs.”

Moreover, according to your suggestion, the title has been revised to highlight the major finding of the paper, which is described as follow:

“Growth of Lithium-Indium Dendrites in High Energy and Power All-Solid-State Lithium Batteries”

According to your suggestion, we have calculated the specific energy and specific power of the cell based on the average discharge voltage (3.1 V), discharge current (3.8 mA cm⁻²), capacity of cathode active material (4 mAh cm⁻²) and cell mass (40+70+25=135 mg). The cell mass excludes current collectors and exterior package, because the cell was tested in the lab without involving commercialized production and elaborated weight optimization. The specific energy and specific power are 72.1 Wh kg⁻¹ and 87.26 W kg⁻¹, respectively. We have summarized the cell-level specific energy and power for the reported typical ASSLBs using powder sulfide electrolytes in our previous work ¹¹. The results is shown as follow (Figure R2). Compared with the existing results, the value of the present work is in the upper right region of Ragone plots, indicating a high energy and power. Currently, the energy and power of ASSLBs are mainly limited by the use of thick electrolyte. The performance can be further improved by reducing the thickness of solid electrolyte in the future.

Figure R2. Ragone plots for cells employing sulfide electrolytes cycled at room temperature. The specific energy and power were delivered during discharging, normalized by the cell mass. ASSLBs with LiCoO_2 (LCO) [X1-X6], $\text{LiNi}_x\text{Co}_y\text{M}_{1-x-y}$ (NCM) [X7-X10] and M_xS ($\text{M} = \text{Co}, \text{Ni}, \text{Ti}, \text{Li}, \text{Fe}$) [X11-X16] are compared. Filled symbol indicates that the cell was charged and discharged at equal current. Empty symbol indicates the cell was charged at a certain current and discharged at different rates. The references X1-X16 can be obtained from our previous work¹¹.

11. Luo S. T. et al. A high energy and power all-solid-state lithium battery enabled by modified sulfide electrolyte film. *J Power Sources* **485**, 229325 (2021).

#Q2.12: There are smaller aspects that should be checked. Spelling mistakes like “columbic efficiency” (should be “Coulomb efficiency”). What is the difference between a “whisker” and a “dendrite”?

#A2.12: Thanks for your suggestions. We have corrected the spelling mistake and carefully checked the full manuscript. The revised Figure 1a is shown as follow:

Figure 1. Cycling performance of the cell LNO@NCM622//LPSCI//Li-In. **a** Long-term cycling performance at 3.8 mA cm⁻² at 25 °C. **b** Galvanostatic charge-discharge profiles from the 890th to the 897th cycle. **c** Galvanostatic charge profile for the 897th cycle.

Regarding “whisker” and “dendrite”, both the two words can be used to describe the morphology of deposited lithium. However, judging from the meaning of the word itself, “whisker” is more focused on sharpness and slenderness, thus it is more appropriate to describe lithium dendrites than Li-In dendrites.

Thank you again for all your suggestions.

For the third reviewer:

“In this paper, the author observed the dendrite growth in Li-In alloys and analyzed its mechanism. The findings on dendrite growth of Li-In contain a certain amount of novelty. However, the reviewer thinks that the Li-In anode is not suitable as an anode material for high energy all-solid-state batteries. In addition, it is stated that the interface between the Li-In alloy and the electrolyte was found to be unstable in this paper, but considering that the Li₆PS₅Cl is unstable under the redox potential of Li-In (0.62 V vs. Li⁺/Li) from theoretical calculation (G. Ceder et al., Chem. Mater. 2016, 28, 266–273), it is obvious that the interface between the Li-In alloy and the electrolyte is unstable. In summary,

although this paper provides new insights into dendrite growth in Li-In, its impact on the development of high-energy all-solid-state batteries is small, and the reviewers recommend that the authors submit this paper to a more specific journal.”

Thanks for your comments. As you mentioned, the Li-In anode is indeed unsuitable for high-energy ASSLBs due to the low capacity and high cost. However, owing to the great mechanical ductility and constant redox potential, Li-In alloy has become one of the most widely used anode in the laboratory for testing the performance of electrolytes or cathodes. Our investigations provide crucial fundamental insights that can help benchmark lab-scale solid-state battery studies, especially at high loading and high current.

Besides, our investigation indicates that excellent cycling and rate performance can be achieved by using alloy anode instead of pure metal Li. Even if Li-In alloy is not utilized, Li-Ag, Li-Al, Li-Mg, Li-Sn and many other alloys still have broad application prospect in the future. However, various elements (including Mg, Ag, Zn, Al, C, Si, Sn, Pb, Sb, and Bi) will undergo different degrees of volume expansion when forming alloy with lithium¹². Therefore, the growth of alloy dendrites is likely to be universal when the cell is cycled at high loading and high current. This finding will provide a rational guidance for the development of high energy and power ASSLBs with alloy anodes. Furthermore, although Li-In anode is unsuitable for large-scale commercial application, it can also be used in some special occasions with its excellent cycling and rate performance, such as medical and aerospace fields.

You mentioned that the interface between the Li-In anode and the electrolyte LPSCl is unstable based on the theoretical calculation. We have carefully read the paper that you recommended. It can be found that the electrochemical window of the electrolyte LPSCl is quite narrow (2.1~2.3 V vs. Li metal). Therefore, the electrolyte may be electrochemically decomposed when contacting with Li-In anode (~0.62 V). However, in the present work, we found that there exist **chemical reaction** between metal In and electrolyte LPSCl, instead of **electrochemical reaction**. According to

the concepts clarified by Banerjee et al.¹³, the chemical reaction is:

“If an electrode and the solid electrolyte have a mismatch of chemical potential, spontaneous chemical reaction(s) may occur once these two materials are put in contact.”

The electrochemical reaction is:

“Most solid electrolytes (SEs) have a narrow electrochemical stability window and cannot operate at the full voltage range of the cathode and anode materials. SEs can be oxidized at high voltages or reduced at low voltages if they have sufficient contact with electronically conductive materials.”

Our experiment results and theoretical calculation demonstrate that InS forms at the LPSCI-In interface. Therefore, it is a chemical reaction involving two materials, not the decomposition of electrolyte itself. This result has never been reported in the previous investigations.

In summary, although it is difficult for Li-In anode to achieve large-scale commercialization, our research is of great significance for benchmarking the lab-scale solid-state battery studies and the development of alloy anodes.

12. Obrovac M.N. et al. Alloy negative electrodes for Li-ion batteries. *Chem. Rev.* **114**, 11444-11502 (2014).

13. Banerjee A. et al. Interfaces and interphases in all-solid-state batteries with inorganic solid electrolytes. *Chem. Rev.* **120**, 6878-6933 (2020).

#Q3.1: Page 3, Line 10

Please write the correct composition of LPSCI when it first appears.

#A3.1: Thanks for your suggestion. We have revised the description in paragraph 3, page 3 as follow:

“Combined with scanning electron microscope (SEM) and scanning transmission electron microscope (STEM) observations, we discovered the growth of Li-In dendrites in **Li₆PS₅Cl (LPSCI)** solid electrolyte, which leads to a rapid capacity fading and subsequent battery failure.”

#Q3.2: Page 3, Line 19-20 and Figure S1

As there are a few peaks that cannot be indexed Li_7PS_6 in Figure S1 around 34° and 47° , the prepared LPSCI is not single phase.

#A3.2: Thanks for your comment. For further verification, we repeated the XRD testing for electrolyte LPSCI. The result is shown as follow. The peak at 34° still exists but quite weak. Compared with the XRD patterns of standard materials, this peak can be indexed to LiCl (PDF#04-0664). The peak at 47° cannot be observed and no materials that possibly existed in the electrolyte match well with it. Therefore, it should be caused by the impurities introduced by the XRD testing. LiCl is one of the raw materials for electrolyte preparation. Although it is difficult to completely remove it, the weak peak demonstrates the ultra-low content of LiCl. Furthermore, as observed from the SEM and EDX mapping of electrolyte LPSCI shown as follow (Figure S2), elements of P, S, and Cl are homogeneously distributed in the LPSCI. Therefore, the composition of the electrolyte is uniform. The ultra-low content of LiCl will not have significant impact on the performance. According to your suggestion, we have revised the description in paragraph 2, page 4 as follow:

“The purity of synthesized LPSCI was determined by XRD patterns (Figure S1 in the supplementary information). The diffraction peaks are well indexed to standard LPSCI. The homogeneity of electrolyte LPSCI is demonstrated by the SEM image and energy-dispersive X-ray spectroscopy (EDX) mapping of P, S and Cl shown in Figure S2.”

Figure S1. XRD patterns of the synthesized LPSCl and standard LPSCl.

Figure S2. SEM image and EDX mapping of P, S, Cl for electrolyte LPSCl.

#Q3.3: Page 4, Line 1-4 and Figure S4

Although the author mentioned that “the formed alloy phase is uniformly distributed in indium matrix”, the cross-sectional SEM image of the Li-In alloy is not uniform.

#A3.3: Thanks for your comment. The Li-In alloy used in the experiment is prepared

by solid-state diffusion. Metal Li and metal In with a weight ratio of 2:98 is combined under a 760 MPa pressure, which is easy-operating and low-costing. Precisely because of this solid-state diffusion method, metal Li and In cannot be absolutely uniformly mixed like smelting method. Fortunately, due to the good diffusivity of lithium in indium, the composition of Li-In anode will become more uniform during the cycling. Besides, we have ever tested the long-term cycling performance of the cell with smelted Li-In alloy. It exhibits a similar dendrite morphology with that of the cell using Li-In anode prepared by solid-state diffusion. Therefore, the growth of Li-In dendrites will not be significantly affected by the initial uniformity of Li-In alloy. According to your suggestion, we have revised the description in paragraph 2, page 4 to be in accord with actual situation.

“Due to the good deformation properties of lithium and indium, Li metal easily diffuses into the In matrix and forms alloy phase under a high pressure.”

#Q3.4: Page 10, Line 1-3 and Figure 4

As mentioned above, the $\text{Li}_6\text{PS}_5\text{Cl}$ is unstable under the redox potential of Li-In (0.62 V vs. Li^+/Li) from theoretical calculation (G. Ceder et al., Chem. Mater. 2016, 28, 266–273), it is normal that the Li-In alloy is unstable towards $\text{Li}_6\text{PS}_5\text{Cl}$ solid electrolyte.

#A3.4: Thanks for your comment. As we have mentioned above, chemical reaction and electrochemical reaction are different. According to the theoretical calculation performed by G. Ceder et al., the electrolyte LPSCl might be electrochemically decomposed at voltage of 0.62 V due to the narrow electrochemical window. However, both experiment results and theoretical calculation in the present work demonstrate that InS forms at the LPSCl-In interface, which belongs to chemical reaction. We also find that the slight interface reaction between In and LPSCl plays an important role for the growth of Li-In dendrites. Both chemical reaction at LPSCl-In interface and the growth of Li-In dendrites have never been reported in the previous investigations.

#Q3.5: Figure 7

The author should reflect on the interfacial side reaction between Li-In alloy and $\text{Li}_6\text{PS}_5\text{Cl}$ into the schematic diagram.

#A3.5: Thanks for your suggestion. We have revised the schematic diagram of Li-In dendrites in Figure 7, which is shown as follow:

Figure 7. Schematic diagram of LPSCl-Li and LPSCl-LiIn interface evolution in cells before and after cycling.

#Q3.6: Page 19, Line 2 and Page20, Line 3

The font of °C is different from the other characters.

#A3.6: Thanks for suggestion. We have corrected the font of °C and carefully checked the full manuscript.

#Q3.7: Page 19, Line 5

Mpa should be revised to be MPa.

#A3.7: Thanks for your suggestion. We have revised all the units “MPa” and carefully checked the full manuscript.

Thank you again for all your suggestions.

REVIEWER COMMENTS

Reviewer #1 (Remarks to the Author):

The authors have addressed all of the concerns and questions that I brought up for the first version of the manuscript. In particular, my biggest concern regarding the In-rich and "regular" Li-In dendrites has been clarified. Therefore, I recommend the present manuscript for publication.

Reviewer #2 (Remarks to the Author):

Authors replied to the referee comments in a good way. Few points are left.

Authors should also still be clearer on how they prepare their Li-In sample. Now they state the used "large pieces" from which a small one is punched out. But this does not seem to appear in the experimental section. An experimental section should contain sufficient information for other experts to reproduce the results, otherwise there is no need for an experimental part.

Authors should also state in the main text the composition of the cathode. How much solid electrolyte was used here? This should be included in the energy calculation.

Authors should also clearly state in the main text how much the theoretical capacity of the Li-In electrode was used per cycle.

Figure R2: Should be updated, e.g. check Ragone plot in <https://onlinelibrary.wiley.com/doi/full/10.1002/aenm.202002394>

Reviewer #3 (Remarks to the Author):

The authors have responded sincerely to the reviewers' comments. However, the reviewer does not still allow the present manuscript for the publication because there are still some questions regarding the following points.

The author described that the chemical reaction was caused by the chemical potential difference between the In metal and LPSCI electrolyte, but since the STEM-EELS observation was performed on the Li-In and LPSCI interface, the chemical potential difference between In and LPSCI cannot be discussed. If the chemical potential difference between the In and LPSCI is the driving force for the formation of the interfacial layer as the authors mentioned, the simply contact between In and LPSCI should result in the interfacial layer formation. The authors should confirm that by using impedance spectroscopy or STEM-EELS, etc.

RESPONSE TO REFEREES

We are very grateful to the reviewers for their thoughtful suggestions and careful reviews on our manuscript. The first reviewer has recommended our manuscript for publication. According to the suggestions of the second and third reviewers, we have revised our manuscript. The detailed changes corresponding to the reviewers' comments are listed below:

For the second reviewer:

Authors replied to the referee comments in a good way. Few points are left.

#Q2.1: Authors should also still be clearer on how they prepare their Li-In sample. Now they state the used "large pieces" from which a small one is punched out. But this does not seem to appear in the experimental section. An experimental section should contain sufficient information for other experts to reproduce the results, otherwise there is no need for an experimental part.

#A2.1: Thanks for your suggestion. According to your suggestion, we have added the detailed information for the preparation of Li-In anode in the experimental section in paragraph 1, page 24, which is described as:

"Lithium-indium (Li-In) alloy was prepared by solid-state diffusion method. A large piece of lithium and a large piece of indium with a weight ratio of 2:98 were pressed together under a high pressure of 760 MPa. Due to the good deformation properties of lithium and indium, Li metal easily diffuses into the In matrix and the Li-In alloy can be obtained. Then the obtained Li-In alloy was cut into disks with a diameter of 10 mm to assemble cells."

#Q2.2: Authors should also state in the main text the composition of the cathode. How much solid electrolyte was used here? This should be included in the energy calculation.

#A2.2: Thanks for your suggestion. According to your suggestion, we have added the composition of the cathode and the weight of electrolyte to ensure the calculation of cell-level energy. The description in paragraph 2, page 24 is revised as follow:

“As-synthesized **LNO@NCM622 powder** was mixed with **LPSCI electrolyte powder** in a mass ratio of **70:30** and the obtained composite was used as cathode for the full cell. Li-In alloy foil was used as anode. Cells were assembled as follow: **70 mg LPSCI electrolyte powder** was pressed under 150 MPa into an electrolyte layer with a diameter of 10 mm. Then, **40 mg composite cathode** powder was uniformly spread on one side of the electrolyte layer and then a stainless steel (SS) foil was put on composite cathode layer. Next, **Li-In alloy (~ 25 mg)** was placed on the other side of the electrolyte layer with a SS foil as collector. Finally, all the components were compressed together under 760 MPa to form full cell.”

#Q2.3: Authors should also clearly state in the main text how much the theoretical capacity of the Li-In electrode was used per cycle.

#A2.3: Thanks for your suggestion. According to your suggestion, we have added the statement of theoretical capacity and the used capacity per cycle for Li-In anode in paragraph 3, page 24, which is described as follow:

“For the assembled cell **LNO@NCM622//LPSCI//Li-In** with a diameter of 10 mm, the **cathode loading is 4 mAh cm⁻²** based on the discharge capacity of the initial cycle. For Li-In anode (25 mg), the metal In and metal Li is 24.5 mg and 0.5 mg, respectively. The **capacity of metal In is 6 mAh cm⁻²** calculated from the theoretical specific capacity of 194 mAh g⁻¹. The metal Li is used to supplement the lithium lost by the cathode during the cycling, thus it is not included in the anode capacity. Therefore, the theoretical capacity of Li-In anode is 6 mAh cm⁻² and higher than that of cathode. Therefore, **67%** theoretical capacity of Li-In anode is used per cycle.”

#Q2.4: Figure R2: Should be updated, e.g. check Ragone plot in <https://onlinelibrary.wiley.com/doi/full/10.1002/aenm.202002394>

#A2.4: Thanks for your suggestion. According to your suggestion, the Ragone plot in

the article that you recommended has been contained in the updated Figure R2, which is indicated by gray star. The revised Figure R2 is shown as follow. Moreover, Figure R2 has been added in the Supplementary information (Fig. 16). The description in the manuscript in paragraph 2, page 18 has been revised as:

“High loading and high current are the necessary conditions for the growth of Li-In dendrites. Energy (Wh kg^{-1}) and power (W kg^{-1}) are positively correlated with the cathode loading and current density, respectively. **Supplementary Fig. 16** shows the Ragone plots for the typical cells employing sulfide electrolytes cycled at room temperature. It can be found that the value of the present work is in the upper right region of Ragone plots, indicating a high energy and power. Therefore, the growth of Li-In dendrites is more likely to occur in high-energy and high-power batteries, which should be given attention in the testing and application of ASSLBs.”

Figure R2. Ragone plots for the cells employing sulfide electrolytes cycled at room temperature. The specific energy and power were delivered during discharging, normalized by the cell mass. ASSLBs with LiCoO_2 (LCO) [X1-X6], $\text{LiNi}_x\text{Co}_y\text{M}_{1-x-y}$ (NCM) [X7-X9] and M_xS ($M = \text{Co, Ni, Ti, Li, Fe, Cu}$) [X10-X15, 2] are compared.

Filled symbol indicates that the cell was charged and discharged at equal current. Empty symbol indicates the cell was charged at a certain current and discharged at different rates. The references X1-X16 can be obtained from our previous work [1].

[1] Luo S. T. et al. A high energy and power all-solid-state lithium battery enabled by modified sulfide electrolyte film. *J Power Sources* **485**, 229325 (2021).

[2] Santhosha A. L. et al. Macroscopic displacement reaction of copper sulfide in lithium solid-state batteries. *Adv. Energy Mater.* **10**, 2002394 (2020).

Thank you again for all your suggestions.

For the third reviewer:

The authors have responded sincerely to the reviewers' comments. However, the reviewer does not still allow the present manuscript for the publication because there are still some questions regarding the following points.

#Q3.1: The author described that the chemical reaction was caused by the chemical potential difference between the In metal and LPSCl electrolyte, but since the STEM-EELS observation was performed on the Li-In and LPSCl interface, the chemical potential difference between In and LPSCl cannot be discussed. If the chemical potential difference between the In and LPSCl is the driving force for the formation of the interfacial layer as the authors mentioned, the simply contact between In and LPSCl should result in the interfacial layer formation. The authors should confirm that by using impedance spectroscopy or STEM-EELS, etc.

#A3.1: Thanks for your comments. You gave us good suggestions that make our investigation more complete. According to your suggestion, a contact experiment was performed with a pure In foil contacted with electrolyte LPSCl for 7 days. Then the electrolyte LPSCl was removed by the above-mentioned washing method. Raman spectroscopy and XPS analysis were conducted to detect the surface of In foil that contacts with LPSCl. Meanwhile, a fresh In foil without contact was also analyzed as a contrast. The results are shown in Fig. 5c and 5d. The detailed results and discussion have been added in paragraph 2, page 16, which is presented as:

Fig. 5 **c** SEM images and Raman spectra of fresh In foil and In foil contacting with LPSCl for 7 days. **d** XPS depth profiling analysis for the In foil contacting with LPSCl for 7 days.

“To further confirm the occurrence of interface reaction and determine the generated product, a contact experiment was performed with a pure In foil contacted with electrolyte LPSCl for 7 days. Then the electrolyte LPSCl was removed by the above-mentioned washing method. Raman spectroscopy and XPS analysis were conducted to detect the surface of In foil that contacts with LPSCl. Meanwhile, a fresh In foil without contact was also analyzed as a contrast. Figure 5c shows the SEM images and Raman spectra of the fresh In foil and contacted In foil. A lot of hillocks appear on the surface of the contacted In foil, which is caused by the intimate combination of electrolyte layer and In foil under high pressure. Comparing the Raman spectra for the surface of In foils, a new peak at 300 cm^{-1} was recognized for the contacted In and it matches well with the Raman spectrum of standard material In_2S_3 , which evidently demonstrates the formation of In_2S_3 . XPS depth analysis for the contacted In foil is shown in Fig. 5d. The In 3d peaks and S 2p peaks of In_2S_3 are detected at the surface of In foil, which is consistent with the results of Raman spectra. As the etching depth increases, the peaks of In 3d change to higher binding energy, indicating a valence transition from In^{3+} to In^0 . Meanwhile, the peaks of S 2p

gradually decrease, suggesting the penetration of interphase layer. As seen from the atomic concentrations of In, S, and Cl from depth profiles in Fig. 5d, the interphase layer is mainly composed of In and S, without significant variation of Cl content detected. Therefore, In_2S_3 is the reaction product between metal In and electrolyte LPSCI.”

Besides, according to your suggestion, we also investigated the influence of interface reaction on cell resistance using electrochemical impedance spectroscopy (EIS) measurements. The results are shown in Supplementary Fig. 13 and the discussion has been added in the manuscript in paragraph 2, page 17, which is presented as follow:

“The influence of interface reaction on cell resistance was investigated by electrochemical impedance spectroscopy (EIS) measurements. Supplementary Fig. 13 shows the evolutions of electrochemical impedance spectrum and corresponding internal resistance of the cell In//LPSCI//In for 144 hours. It can be clearly found that the internal resistance first decreases and then keep stable. The reduction of internal resistance is attributed to the improved interface contact property caused by the interface reaction. The interphase layer has a good wettability with electrolyte, which can be reflected from Fig. 4b and 4c. The subsequently unchanged resistance represents the formation of stable interphase layer. Therefore, the interface reaction between In and LPSCI will not induce increased internal resistance, which agrees well with the great cycling performance shown in Fig. 1a.”

Supplementary Fig. 13 Evolutions of electrochemical impedance spectrum and

corresponding internal resistance of the cell In//LPSCI//In for one week.

Thank you again for your suggestions.

REVIEWERS' COMMENTS

Reviewer #3 (Remarks to the Author):

The authors have responded sincerely to the reviewers' comments. The reviewer does not have further comment before publication.